# RNA-binding protein YebC enhances translation of proline-rich amino acid stretches in bacteria

Dmitriy Ignatov[1], Vivekanandan Shanmuganathan[1,4], Rina Ahmed-Begrich [1,4], Kathirvel Alagesan[1], Karin Hahnke[1], Chu Wang[1], Kathrin Krause[1], Fabián A. Cornejo [1], Kristin Funke [2], Marc Erhardt [1,2], Christian Karl Frese[1,3] & Emmanuelle Charpentier [1,2] ✉

The ribosome employs a set of highly conserved translation factors to efficiently synthesise proteins. Some translation factors interact with the ribosome in a transient manner and are thus challenging to identify. However, proteins involved in translation can be specifically identified by their interaction with ribosomal RNAs. Using a combination of proteomics approaches, we identified 30 previously uncharacterized RNA-binding proteins in the pathogenic bacterium *Streptococcus pyogenes*. One of these, a widely conserved protein YebC, was shown to transiently interact with 23S rRNA near the peptidyl-transferase centre. Deletion of *yebC* moderately affected the physiology and virulence of *S. pyogenes*. We performed ribosome profiling and detected increased pausing at proline-rich amino acid motifs in the absence of functional YebC. Further experiments in *S. pyogenes* and *Salmonella* Typhimurium and using an in vitro translation system suggested that YebC is a translation factor required for efficient translation of proteins with proline-rich motifs.

Bacteria use RNA-binding proteins (RBPs) to express and regulate genes[1]. Despite decades of studies on bacterial genetics and physiology, many proteins still have unknown functions. Some of these proteins might perform their function by interacting with RNA. Recently, several methods have been developed to identify bacterial RBPs on a whole-proteome scale. The Grad-seq approach identifies RBPs that form stable complexes with RNA by the transcriptomic and proteomic analysis of size-fractionated cellular lysates[2]. More transient protein-RNA interactions can be captured by UV crosslinking. The resulting covalently bound protein-RNA complexes can be specifically enriched using purification on silica beads[3], organic extraction[4–6], or pull-down of artificially polyadenylated bacterial transcripts[7]. Subsequent proteomic analysis of the proteins cross-linked with RNA revealed a repertoire of RBPs in *Escherichia coli*[3,4,6,7], *Salmonella* Typhimurium[5]

and *Staphylococcus aureus*[8]. These studies identified a novel regulatory protein, ProQ[2] and showed that the glycolytic enzyme enolase in *E. coli* and the transcription factor CcpA in *S. aureus* function as RBPs[3,8]. Several proteins of unknown function have also been shown to interact with RNA[5–7]; further characterisation of these proteins might uncover new aspects of prokaryotic RNA biology.

*Streptococcus pyogenes* is a Gram-positive bacterial pathogen causing not only benign human infections such as pharyngitis and impetigo, but also potentially life-threatening invasive diseases such as septicaemia, streptococcal toxic shock syndrome and necrotising fasciitis[9]. *S. pyogenes* has served as a model organism for RNA biology: the transcriptome analysis has identified multiple non-coding RNAs[10], the analysis of RNA cleavage by different ribonucleases has provided insights into their modes of action[11] and the study of the CRISPR-Cas9

[1]Max Planck Unit for the Science of Pathogens, Berlin, Germany. [2]Institute of Biology, Humboldt-Universität zu Berlin, Berlin, Germany. [3]Present address: Bayer AG, Wuppertal, Germany. [4]These authors contributed equally: Vivekanandan Shanmuganathan, Rina Ahmed-Begrich. ✉e-mail: research@emmanuelle-charpentier.org

system has resulted in the development of new biotechnological tools[12]. In this study, we systematically characterised RBPs in *S. pyogenes*. Our approach detected most of the annotated RBPs involved in RNA synthesis and degradation, as well as ribosome assembly and translation. In addition, we identified a group of proteins of unknown function that interact with RNA. One of these proteins, YebC, was selected for further characterisation.

YebC is a widely conserved bacterial protein. A recent study has shown that YebC interacts with RNA in *E. coli*, which corroborates our data[6]. The mitochondrial homologues of YebC, TACO1 in mice and DPC29 in *Saccharomyces cerevisiae*, have been shown to associate with the mitochondrial ribosome. In humans and mice, TACO1 is required for efficient translation of the *COXI* mRNA and its absence results in deficiency of respiratory complex IV[13,14]. Recently, TACO1 was shown to be required for efficient translation of the polyproline motifs in mitochondria. The amino acid sequence of COXI contains polyproline motifs. In the absence of TACO1, the mitoribosome stalls at these polyproline motifs and cannot efficiently synthesise the COXI protein[15].

In bacteria, translation elongation factor P (EF-P) is required for efficient translation of proteins containing polyproline motifs. The consecutive proline residues impose a peptidyl-tRNA geometry that is unfavourable to the translating ribosome. In the absence of EF-P, the ribosome stalls at polyproline motifs. The interaction of EF-P with the ribosome promotes a geometry favourable for the formation of peptide bonds and alleviates the translational stalling[16–18]. Recently, it has been shown that an ABCF ATPase YfmR/Uup also promotes translation of proline-rich motifs[19–21]. Two studies performed in bacteria suggest a role for YebC in translation. In *E. coli*, overexpression of YebC rescued the growth defect caused by inactivation of the ribosome-associated GTPase BipA[22]. In *B. subtilis*, overexpression of YebC2 (YeeI) rescued the defect of flagellar assembly caused by deletion of the translation EF-P. Cells lacking EF-P were defective in hook completion due to translation pausing at a specific PP motif within the flagella protein FliY[23]. While these studies suggest that YebC is involved in translation and is probably required for the translation of polyproline motifs, another group of studies suggest that the protein may function as a transcription factor[24–28]. Our study shows that YebC enhances the translation of polyproline motifs both in vivo and in vitro, and thus represents a translation factor.

## Results

### UV-mediated RNA-protein crosslinking identifies RBPs in *S. pyogenes*

The initial step in our study was identification of RBPs in *S. pyogenes*. To discover proteins interacting with RNA, we grew *S. pyogenes* in a chemically-defined medium (CDM) to mid-logarithmic phase and irradiated the cells with UV light at 254 nm. The cross-linked protein-RNA complexes were specifically enriched using the orthogonal organic phase separation (OOPS) technique[4] (Fig. 1a and Supplementary Fig. 1). *S. pyogenes* is a natural auxotroph for lysine and arginine[29], allowing the use of stable isotope labelling by amino acids in cell culture (SILAC)[30] to calculate the abundance of proteins in the UV irradiated relative to non-irradiated samples. The resulting OOPS enrichment values were used to measure the interaction with RNA (Supplementary Data 1). We further identified the amino acids cross-linked to RNA using the RBS-ID approach (Supplementary Fig. 2)[31]. In total, we identified 1440 RNA cross-linking sites across 254 *S. pyogenes* proteins. The identified RNA cross-linking sites were often located in domains of ribosomal proteins and in RNA-binding domains. Next, we searched the KEGG and Gene Ontology databases to create a list of annotated RBPs for *S. pyogenes*[32,33]. After manual curation, 182 *S. pyogenes* proteins were annotated as RBPs (Supplementary Data 2). Most of the annotated RBPs were enriched by OOPS and contained RNA

cross-linking sites (Fig. 1b). These include RBPs involved in both RNA metabolism and translation (Fig. 1c).

### Proteins of unknown function interact with RNA

We hypothesised that some of the OOPS-enriched proteins not annotated as RBPs represent novel RBPs. 59 proteins not annotated as RBPs showed statistically significant OOPS enrichment and contained RNA cross-linking sites (Fig. 1b). We narrowed this selection to 30 proteins using an OOPS enrichment greater than two as an additional criterion (Supplementary Fig. 3), and considered these proteins as potential RBP candidates (Table 1). Some of them have been identified as novel RBPs in previous studies aimed at global identification of RBPs in bacteria[3–8].

Seven of these potential RBPs are annotated as DNA-binding proteins (Fig. 2a). One of them, the transcription factor CcpA, a master regulator of carbon metabolism, has recently been shown to function as a non-canonical RBP in *S. aureus*[8]. The other RBP candidates are several metabolic enzymes and proteins involved in transport that could moonlight as RBPs. One signalling protein, the small alarmone synthase RelQ, has previously been shown to interact with RNA in vitro[34]. The ribosome-associated proteins, peptide deformylase and trigger factor, have been shown to interact with 23S rRNA[35,36].

Ten RBP candidates are of unknown function. We selected five of them and investigated their cross-linking with RNA using immunoprecipitation and radioactive labelling of co-precipitated RNA with T4 polynucleotide kinase (PNK) (Fig. 2b, c)[37]. To this end, we introduced 3x FLAG tags to the C-termini of their genes. Expression from the native promoters enabled us to estimate the interaction with RNA at physiological concentrations of the proteins. The positive control gene *yhaM* and the negative control gene *gapN* were also tagged with a 3x FLAG tag. The exoribonuclease YhaM has been shown to form stable complexes with RNA in *Streptococcus pneumoniae*[38]. Consistent with this, we observed the radioactive signal originating from the co-immunoprecipitated RNA only upon UV cross-linking and in the presence of PNK. For the negative control protein GapN, we also observed a signal in the UV + PNK+ sample, but it was only marginally higher than in the UV − PNK+ sample. Therefore, a strong increase in radioactive signal after UV cross-linking and in the presence of PNK indicates that a protein interacts with RNA. The proteins YebC and YjbK immunoprecipitated in amounts comparable with the control proteins. In the UV irradiated and PNK labelled samples, they showed radioactive signals comparable to those of our positive control YhaM, suggesting that YebC and YjbK are bona fide RBPs. YgaC and ThuC, two other proteins of unknown function, immunoprecipitated in low amounts under the conditions tested. However, specific radioactive signals from the cross-linked RNA were also observed for these proteins. Another RBP candidate, the protein PhoH showed weak cross-linking with RNA. These results validate our approach to the search for previously uncharacterised RBPs and show that most of them interact with RNA.

YebC is a broadly conserved protein, suggesting that it plays an important role in bacterial physiology. Our results showed that this protein interacts with RNA, and further study of this protein could uncover a novel molecular biology mechanism. In bacteria, YebC homologues are divided into two subtypes: YebC_I and YebC_II[39]. Some bacteria, such as *E. coli* and *B. subtilis*, encode both subtypes, while *S. pyogenes* only possesses YebC_II (Fig. 2d). The sequence and structure of YebC are evolutionarily conserved (Supplementary Fig. 4). We modelled the structure of YebC from *S. pyogenes* based on YebC from *Coxiella burnetii*[40]. The protein consists of three domains: domains I and II contain the positively charged surface patches, while domain III is highly negatively charged (Fig. 2e). Interestingly, the RNA cross-linking sites identified in our experiment are located near the positively charged surfaces, suggesting that they form the interface for interaction with RNA.

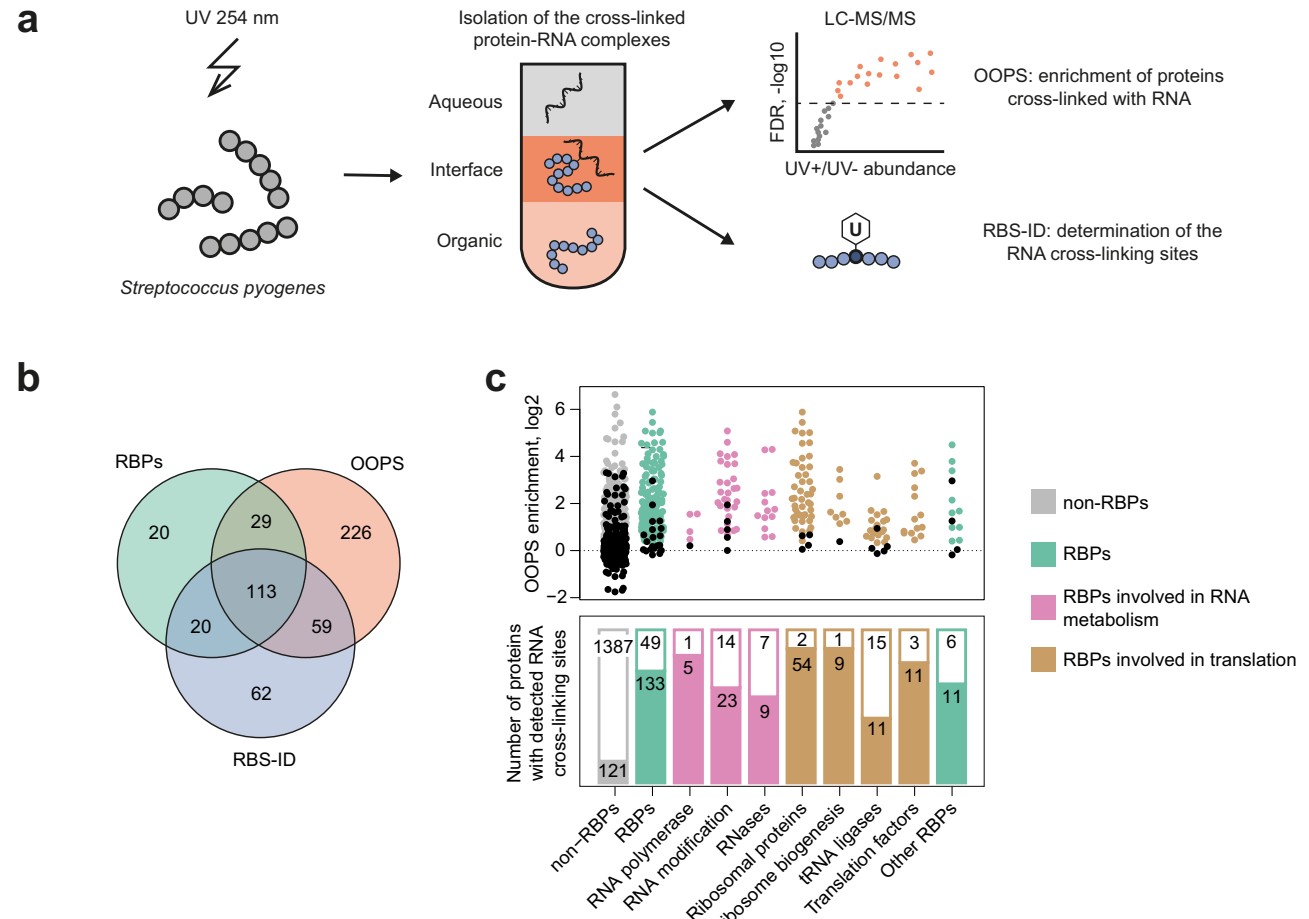

**Fig. 1 | Identification of RBPs in *S. pyogenes*. a** Workflow for the characterisation of proteins that cross-link with RNA. After UV 254 nm cross-linking, the protein-RNA complexes were separated from proteins and RNA by acid guanidinium thiocyanate-phenol-chloroform extraction. The cross-linked protein-RNA complexes were further analysed by OOPS or RBS-ID approaches. **b** Overlap between annotated RBPs, proteins with statistically significant OOPS enrichment and proteins with RNA cross-linking sites identified by RBS-ID. **c** OOPS enrichment and the presence of RNA cross-linking sites for different functional groups of the annotated RBPs. The OOPS enrichment values for proteins in each manually curated category are presented in a bee swarm plot. The proteins whose OOPS enrichment is not statistically significant are indicated by black dots. The bar plot shows the number of proteins with identified RNA cross-linking sites (coloured bars) and without them (white bars) for each category.

## Deletion and site-directed mutagenesis of YebC in *S. pyogenes*

We constructed a strain of *S. pyogenes* with a deleted *yebC* gene and a complemented strain in which *yebC* with a C-terminal 3x FLAG tag was reintroduced on an integrative vector under the control of its native promoter. The deletion of *yebC* slightly affected *S. pyogenes* growth: while no difference was observed in THY or C medium, the Δ*yebC* strain grew to a higher optical density in CDM (Fig. 3a). The difference in growth observed between the media cannot be explained by the differences in YebC expression, since the protein is expressed at similar levels in THY and CDM (Supplementary Fig. 5a). Cell size is also not affected by the *yebC* deletion (Supplementary Fig. 5b, c).

The transcriptome analysis performed in THY medium revealed a moderate effect of *yebC* deletion on gene expression (Supplementary Data 3 and Supplementary Fig. 6a). Notably, the deletion of *yebC* significantly decreased the expression of the major virulence factor of *S. pyogenes*, the secreted protease SpeB[9,41] (Fig. 3b and Supplementary Fig. 6b). We wanted to understand the mechanism of the observed *speB* down-regulation. The expression of *speB* from a heterologous promoter demonstrated that its translation and secretion are not affected by the deletion of *yebC* (Fig. 3b). In contrast, the activity of the *speB* promoter is reduced in the Δ*yebC* strain (Fig. 3c). The deletion of *yebC* also led to reduced intracellular survival in human macrophages (Fig. 3d). In addition, while no difference in *S. pyogenes*-induced

cytotoxicity could be observed between the WT, Δ*yebC* or Δ*yebC/yebC+* strains (Supplementary Fig. 7a), there was a mild albeit not significant reduction of IL-1β and IL-18 secretion by macrophages infected with the Δ*yebC* strain (Supplementary Fig. 7b). In contrast, levels of IL-6 remained unaffected.

Next, we used site-directed mutagenesis to identify amino acid residues critical for YebC function. Several groups of conserved residues forming positive and negative patches on the protein surface were selected (Fig. 4a) and substituted with alanine. The mutated versions of *yebC* were introduced into the Δ*yebC* strain. The deletion of *yebC* resulted in an increase in the optical density of the stationary phase culture in CDM (Fig. 4b) and decrease in the level of SpeB in the culture supernatant (Fig. 4c). Complementation with the wild type *yebC* restored both phenotypes. Some of the mutated variants of *yebC* could not restore the phenotypes, suggesting that these mutations inactivate the protein.

The M1 mutation is a substitution of the highly conserved K21 and K25 with alanine residues. This mutation significantly reduced the level of YebC, probably affecting the protein's stability. Nevertheless, the M1 variant was able to fully complement the Δ*yebC* phenotype, showing that the M1 mutation does not inactivate the protein. The M2 mutation comprises several conserved amino acids that form two positively charged patches on the surface of YebC. This mutation makes *yebC*

**Table 1 | Candidates for novel RBPs in *S. pyogenes***

| Locus tag | Gene | UniProt | Function | OOPS score mid-log, log2 | Ortholog identified as RBP |
|---|---|---|---|---|---|
| SPy_0190 | - | Q9A1M3 | DNA-binding | 2.182 | - |
| SPy_0316* | yebC | P67188 | Unknown | 1.199 | refs. 3–7 |
| SPy_0471* | phoH | Q9A145 | Unknown | 1.772 | refs. 3,4,6,7 |
| SPy_0514 | ccpA | Q9A118 | DNA-binding | 1.307 | refs. 3,5,8 |
| SPy_0534 | aroE.2 | Q9A102 | Metabolic enzyme | 1.369 | - |
| SPy_0539* | thuC | Q9A0Z7 | Unknown | 1.415 | - |
| SPy_0598 | - | Q9A0V5 | Metabolic enzyme | 1.868 | - |
| SPy_0714 | adcA | Q9A0L9 | Transport | 1.673 | ref. 8 |
| SPy_0752 | - | Q9A0J4 | Metabolic enzyme | 1.514 | - |
| SPy_0903 | tcyJ | Q9A074 | Transport | 1.323 | refs. 3–5 |
| SPy_0939 | - | Q9A042 | DNA-binding | 1.098 | - |
| SPy_1121 | - | Q99ZR2 | Unknown | 2.145 | - |
| SPy_1124* | yjbK | Q99ZQ9 | Unknown | 2.386 | - |
| SPy_1125 | relQ | Q99ZQ8 | Signalling | 1.529 | ref. 34 |
| SPy_1126 | nadK | P65781 | Metabolic enzyme | 3.689 | - |
| SPy_1193 | - | Q99ZK4 | Unknown | 1.640 | - |
| SPy_1259 | - | Q99ZE8 | DNA-binding | 1.607 | - |
| SPy_1429 | gpmA | Q99Z29 | Metabolic enzyme | 1.168 | refs. 4,6 |
| SPy_1534 | - | Q99YU4 | Unknown | 2.608 | - |
| SPy_1582 | - | Q99YR0 | Unknown | 2.835 | - |
| SPy_1607 | recX | Q99YP2 | DNA-binding | 1.999 | ref. 8 |
| SPy_1608* | ygaC | Q99YP1 | Unknown | 3.211 | - |
| SPy_1744 | accD | Q99YE0 | Metabolic enzyme | 1.764 | ref. 8 |
| SPy_1746 | fabZ | P64110 | Metabolic enzyme | 1.220 | refs. 6,8 |
| SPy_1751 | fabK | Q99YD4 | Metabolic enzyme | 1.537 | ref. 8 |
| SPy_1862 | ybaB | P67268 | DNA-binding | 2.267 | ref. 5 |
| SPy_1896 | tig | P0C0E1 | Protein metabolism | 1.359 | refs. 3,4,6,7,36 |
| SPy_1936 | degV | P67374 | Unknown | 1.082 | - |
| SPy_1958 | def | P68771 | Protein metabolism | 1.284 | refs. 3,35 |
| SPy_1960 | - | Q99XY5 | DNA-binding | 1.884 | - |

For each RBP candidate, a locus tag, a gene name (if available), a UniProt ID, a manually curated functional annotation and OOPS enrichment values are provided. If orthologs of the proteins have previously been identified as RBPs in other bacteria, the references for these studies are provided. The asterisks indicate the proteins whose interaction with RNA was examined using immunoprecipitation and radioactive labelling of the co-precipitated RNA.

unable to restore the phenotype. With the M3 and M4 mutations, we separately substituted the groups of amino acids that form these positively charged patches. These mutant versions of *yebC* also could not complement the phenotype, indicating that the positively charged amino acids located on the upper part of the protein are essential for its function. The *yebC* gene with the mutation M5 also did not complement the phenotype, suggesting the importance of the conserved amino acids forming the negatively charged surface at the lower tip of domain III. In contrast, the M6 mutation, which affects several negatively charged amino acids in domain II, had no effect on YebC activity. Amino acids Y84 and E85 are universally conserved and Y84 represents

an RNA cross-linking site (Fig. 2e). The *yebC* variant with tyrosine 84 substituted by alanine, but not by phenylalanine, failed to restore the phenotype, suggesting that the tyrosine's aromatic ring, but not its hydroxyl group, is essential for YebC activity. Substitution of glutamic acid 85 by alanine had no effect on YebC activity and this amino acid residue appears to be dispensable.

Finally, we used the OOPS approach to test whether the M2 and Y84A mutations affect the interaction of YebC with RNA (Fig. 4d). The amounts of the mutant versions of YebC were significantly reduced in the RBP-enriched fraction, suggesting that these mutations impair the interaction with RNA. In conclusion, the site-directed mutagenesis revealed the importance of the positively charged amino acids at the upper surfaces of domains I and II, and of the universally conserved Y84 for YebC activity.

## YebC transiently interacts with the ribosome

Next, we used the iCLIP approach to study the interaction of YebC with RNA. The YebC variant with the C-terminal 3x FLAG tag was able to fully complement the Δ*yebC* phenotype (Fig. 4b, c), suggesting that the tag does not interfere with YebC function. We irradiated the *yebC::3xFLAG* strain with UV 254 nm light. The cross-linked YebC-RNA complexes were immunoprecipitated, the RNA was partially degraded with RNase I, resolved on the gel and transferred to the membrane (Supplementary Fig. 8). The radioactively labelled RNA fragments were excised from the membrane and cDNA libraries were prepared according to the iCLIP protocol[42]. In parallel, the control libraries were prepared from UV irradiated WT and non-irradiated *yebC::3xFLAG* strains. As another control for non-specific background, we prepared the size-matched input (SMI) libraries: the lysates of the UV-irradiated *yebC::3xFLAG* strains were resolved on a gel in parallel with other samples, and RNA fragments of the same length were excised from the membrane and used to prepare cDNA libraries[43]. iCLIP relies on reverse transcriptase stopping at the cross-link site between the RNA and peptides remaining after protein digestion. To analyse the iCLIP data, we extracted the cross-linked nucleotides located immediately upstream of the aligned cDNA reads[44] and identified the regions enriched with cross-linked nucleotides (Supplementary Data 4).

More than 20% of the cross-linked nucleotides in the *yebC::3xFLAG* UV+ sample were located in a single region of 23S rRNA (Fig. 5a). No other region in this sample or in any of the controls contained such a high proportion of the cross-linked nucleotides. This locus of 23S rRNA was enriched for the cross-linked nucleotides only in the *yebC::3xFLAG* UV+ sample and not in the controls (Fig. 5b), suggesting that it represents the binding site for YebC. Visual inspection of the cross-linked nucleotides and regions in the genome browser did not identify any other locus strongly enriched in *yebC::3xFLAG* UV+ compared with the controls. Therefore, YebC likely represents a protein that interacts specifically with 23S rRNA. The cDNA fragments mapping to this region encompass helix 89, and most of them terminate directly upstream of nucleotides A2453, A2454 and A2456, which apparently represent the cross-linked nucleotides (Fig. 5c). These nucleotides are homologous to A2450, A2451 and A2453 in *E. coli* (Fig. 5d). They are part of the peptidyl-transferase centre (PTC) and are essential for ribosome function[45].

The observed cross-linking of YebC to the 23S rRNA suggests that this protein could be involved in ribosome biogenesis or translation. We resolved ribosomes isolated from mid-logarithmic and stationary growth phase cultures on sucrose density gradients and demonstrated that YebC does not form a stable complex with translating ribosomes or free ribosomal subunits (Supplementary Fig. 9a). Deletion of *yebC* did not affect the polysome profile (Fig. 5e), suggesting that the protein is likely not involved in ribosome maturation. In conclusion, our data indicate that YebC transiently interacts with the ribosome at or near the PTC and Helix 89 and therefore may affect the peptide bond formation.

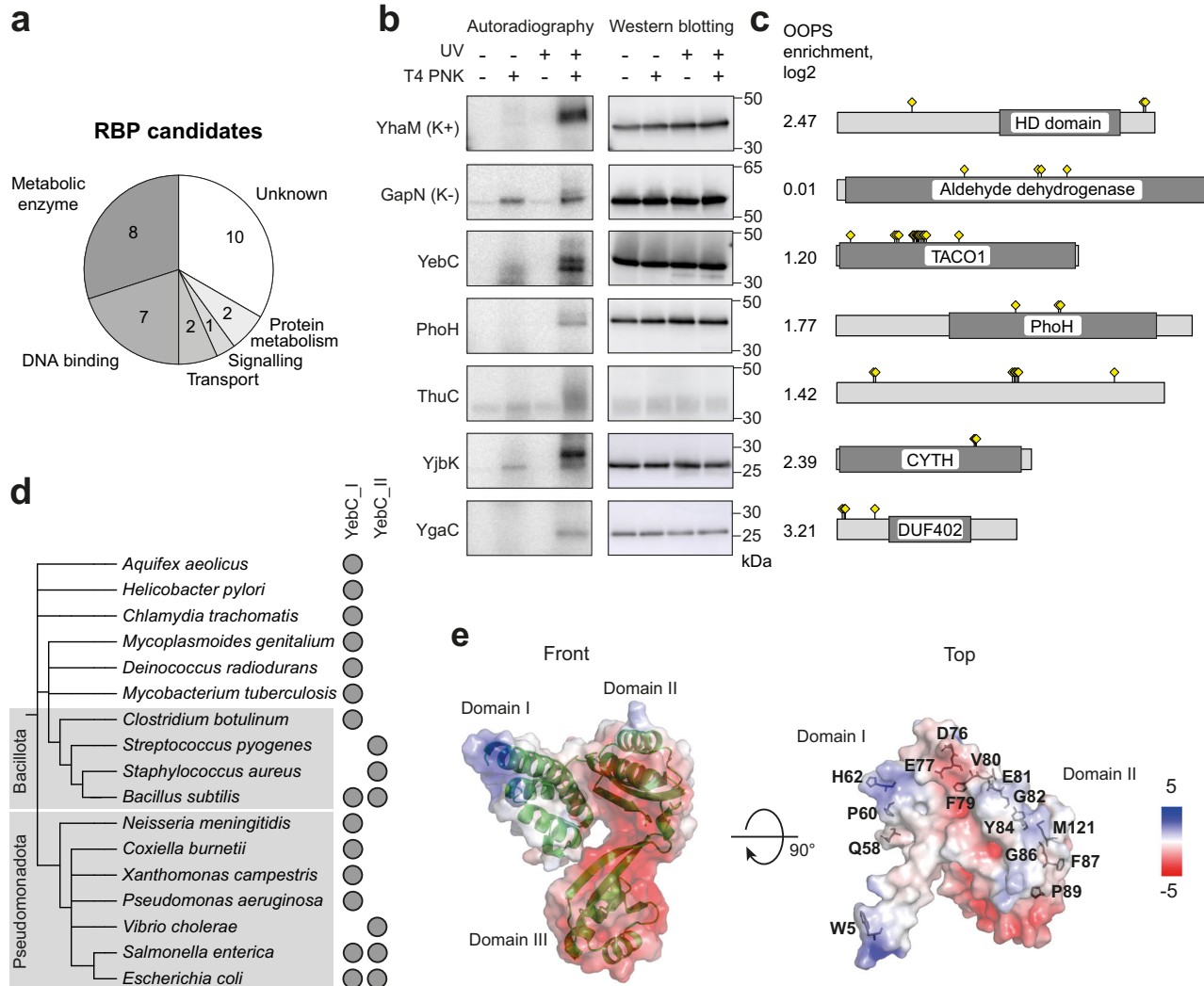

**Fig. 2 | RBP candidates in *S. pyogenes*. a** Manually curated functional categories of RBP candidates. **b** Detection of cross-linked protein-RNA complexes. The strains containing the 3x FLAG tagged potential RBPs or the control proteins YhaM and GapN were UV irradiated. The protein-RNA complexes were immunoprecipitated, radioactively labelled with T4 PNK, separated by gel electrophoresis and transferred to a membrane. Radioactive signals were detected by phosphor imaging. Western blotting with an anti-FLAG antibody served as a control for successful immunoprecipitation. The experiment was performed in two biological replicates with similar results. **c** Description of proteins in panel **b**. OOPS enrichment, annotated domains and location of RNA cross-linking sites are shown. **d** Presence of YebC_I and YebC_II subtypes in different bacteria. **e** Modelled structure of *S. pyogenes* YebC. The cartoon structure of the protein and the predicted surface electrostatic potential are shown. The identified RNA cross-linking sites are indicated in the top view of the protein.

## In the absence of YebC, the ribosome pauses at proline-rich motifs

To study how the mutation of *yebC* affects translation, we used the ribosome profiling approach. The strains with functional YebC (*WT* and Δ*yebC/yebC*+) and with *yebC* mutation (Δ*yebC* and Δ*yebC/yebC_M2*) were grown to the mid-logarithmic growth phase and collected by rapid filtration to preserve the native positions of the translating ribosomes[46]. To accurately identify the ribosome positions, we selected the 27–40 nucleotide long ribosome-protected mRNA fragments and mapped the ribosomal P sites 15 nucleotides upstream of the 3′ ends of these mRNA fragments (Supplementary Fig. 10a, b)[47]. For each codon, we calculated the pause score, defined as the number of reads mapped to a particular position divided by the average read density for the corresponding gene. Median pause scores were slightly higher when serine or glycine codons were located in the E site (Supplementary Fig. 10c). This amino acid specific increase in pausing is explained by the use of chloramphenicol to stop translation in the lysate[48] and does not depend on the presence of functional YebC.

The ribosome pause scores of the Δ*yebC* and Δ*yebC/yebC_M2* strains were similar, but different from those in strains with functional YebC, further demonstrating that the M2 mutation inactivates YebC (Supplementary Fig. 11). We searched for codons where the ribosome pausing changes upon YebC mutation. In total, we identified 168 positions with a statistically significant difference in pause score (Supplementary Data 5). For 153 of these positions, the ribosome pausing was increased in the strains with inactive YebC. Interestingly, pausing was strongly increased in the vicinity of a triple proline codon in the *valS* mRNA (Fig. 6a). Analysis of the genomic context demonstrated that the ribosome pausing was often increased directly at and downstream of proline-rich amino acid sequences (Fig. 6b). The codons located in close proximity to each other were often affected, and we identified a total of 81 positions across the transcriptome showing increased pausing in *yebC* mutants. Most of the loci were located in the vicinity of the PP, PXP or DXP amino acid motifs. Among these, the sequences PPG, PIP and DIP were overrepresented (Fig. 6c and Supplementary Data 6). We searched for these amino acid sequences in *S. pyogenes* proteins and calculated the average pause

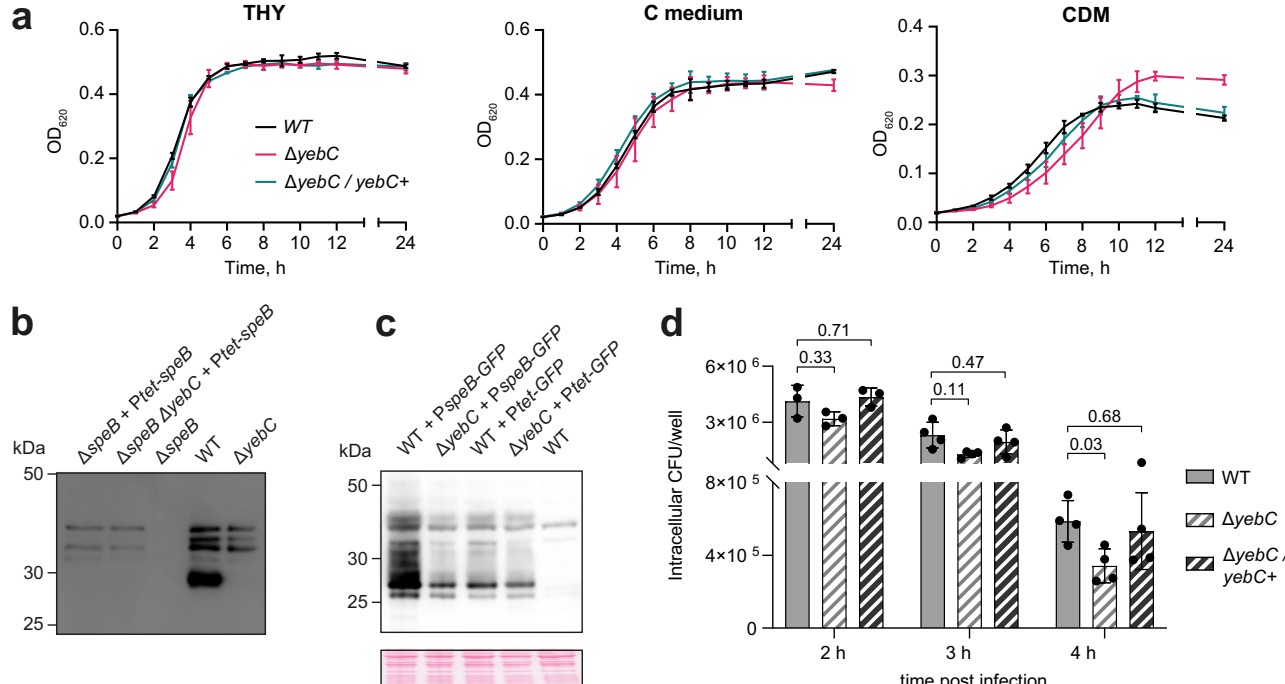

**Fig. 3 | Phenotype of *yebC* deletion in *S. pyogenes*. a** Growth curves of the WT, Δ*yebC* and Δ*yebC*/*yebC*+ strains in THY, C medium and CDM. Data represent the mean ± standard deviation (SD) of three biological replicates. **b** Expression of SpeB in the reporter strains Δ*speB* + P$_{tet}$-*speB* and Δ*speB* Δ*yebC* + P$_{tet}$-*speB*, and in the WT and Δ*yebC* strains. The strains were grown in THY until the early stationary growth phase, the supernatants were collected and SpeB expression was probed with anti-SpeB antibodies. To induce the *tet* promoter, 10 ng/mL anhydrotetracycline was added to the cultures of the reporter strains. In the culture supernatant of the WT strain, SpeB was mostly present as a mature 28 kDa enzyme, while in the supernatants of the *yebC* mutants, only the bands corresponding to the 40 kDa zymogen and several intermediates were detected. The experiment was performed in three biological replicates with similar results. **c** Activity of *speB* promoter in the WT and Δ*yebC* strains. The reporter strains expressing sfGFP under the control of *speB* or *tet* promoters were grown in THY with 10 ng/mL anhydrotetracycline until the early stationary growth phase. The cells were lysed and sfGFP (27 kDa) expression was analysed by western blotting. The Ponceau staining of the membrane demonstrates equal sample loading. The experiment was performed in three biological replicates with similar results. **d** Intracellular survival of *S. pyogenes* WT, Δ*yebC* and Δ*yebC*/*yebC*+ in primary human macrophages at 2 h, 3 h and 4 h post-infection. Data represent the mean ± SD of three (2 h) or four (3 and 4 h) biological replicates. Statistical analysis was performed using two-way ANOVA and the *p* values are indicated.

score in their vicinity (Fig. 6d). In the Δ*yebC* strain, the pause score at PPG increases when the second proline is located in the P site and remains elevated for three codons after PPG translation. The increase in pause score for PIP and DIP is not as pronounced, suggesting that these motifs induce a weaker ribosome pausing. However, in this case, pausing continues even after the ribosome has translated several codons beyond the motifs.

### YebC rescues ribosome stalling at proline-rich motifs

Our ribosome profiling data suggest that YebC alleviates ribosome stalling during translation of the proline-rich amino acid motifs. We examined how this affects the expression of polypeptides with the polyproline sequences. To this end, we constructed the N-terminal 3x FLAG-tagged sfGFP-mKate fusions with different proline-rich sequences in the linker and introduced them to the WT and Δ*yebC* strains of *S. pyogenes*. Expression of the reporter proteins was induced for a short period of time and analysed by western blotting with anti-FLAG antibodies (Fig. 7a). In the WT strain, the reporter protein expression was not affected by the introduction of the proline-rich sequences. In contrast, expression of the reporter protein with P5 and P3 motifs was significantly downregulated in the Δ*yebC* strain. The PPG linker also caused a slight down-regulation, which was however not statistically significant, and the PIP linker had no effect. Interestingly, in the Δ*yebC* strain, we also observed reporters with P5, P3 and PPG linkers that were truncated at the proline-rich stretches. The truncated products probably appear when the ribosome stalls at these amino acid stretches and is unable to synthesise the full-length proteins.

This experiment suggested that YebC is essential for efficient translation of proteins containing polyproline stretches. We wanted to examine how the increased ribosome pausing in the absence of YebC affects protein expression at the whole-proteome level. To this end, we performed a proteome analysis of the same bacterial cultures that we used for the transcriptome analysis: WT, Δ*yebC* and Δ*yebC* / *yebC*+ strains in mid-logarithmic and stationary growth phases (Supplementary Data 7). In general, deletion of *yebC* had a very limited effect on the proteome: only a few genes were differentially expressed between the WT and Δ*yebC* strains (Supplementary Fig. 12). Interestingly, the proteins with YebC-dependent ribosome pause sites demonstrated a slight but statistically significant down-regulation in the Δ*yebC* strain in the logarithmic growth phase (Fig. 7b). In the stationary growth phase, we also observed a trend towards down-regulation, but it was not statistically significant.

We next used PURE in vitro translation system[49] to test whether YebC can rescue ribosome stalling at polyproline amino acid stretches in vitro. This is a reconstituted system in which all components are purified from *E. coli*. Unlike *S. pyogenes*, the genome of *E. coli* encodes two subtypes of *yebC* (Fig. 2d). To investigate the differences between the subtypes, we expressed and purified *E. coli* proteins YeeN (YebC_II) and YebC (YebC_I). We also expressed and purified *E. coli* YeeN with the M2 and Y84A mutations, which, as previously demonstrated in this study, inactivate the protein's activity in *S. pyogenes*. As templates for in vitro translation, we used mRNAs encoding the mutant versions of the FolA protein with an N-terminal FLAG tag (Fig. 8a). Translation of the P0 variant produced a full-length protein of 18.9 kDa. In contrast,

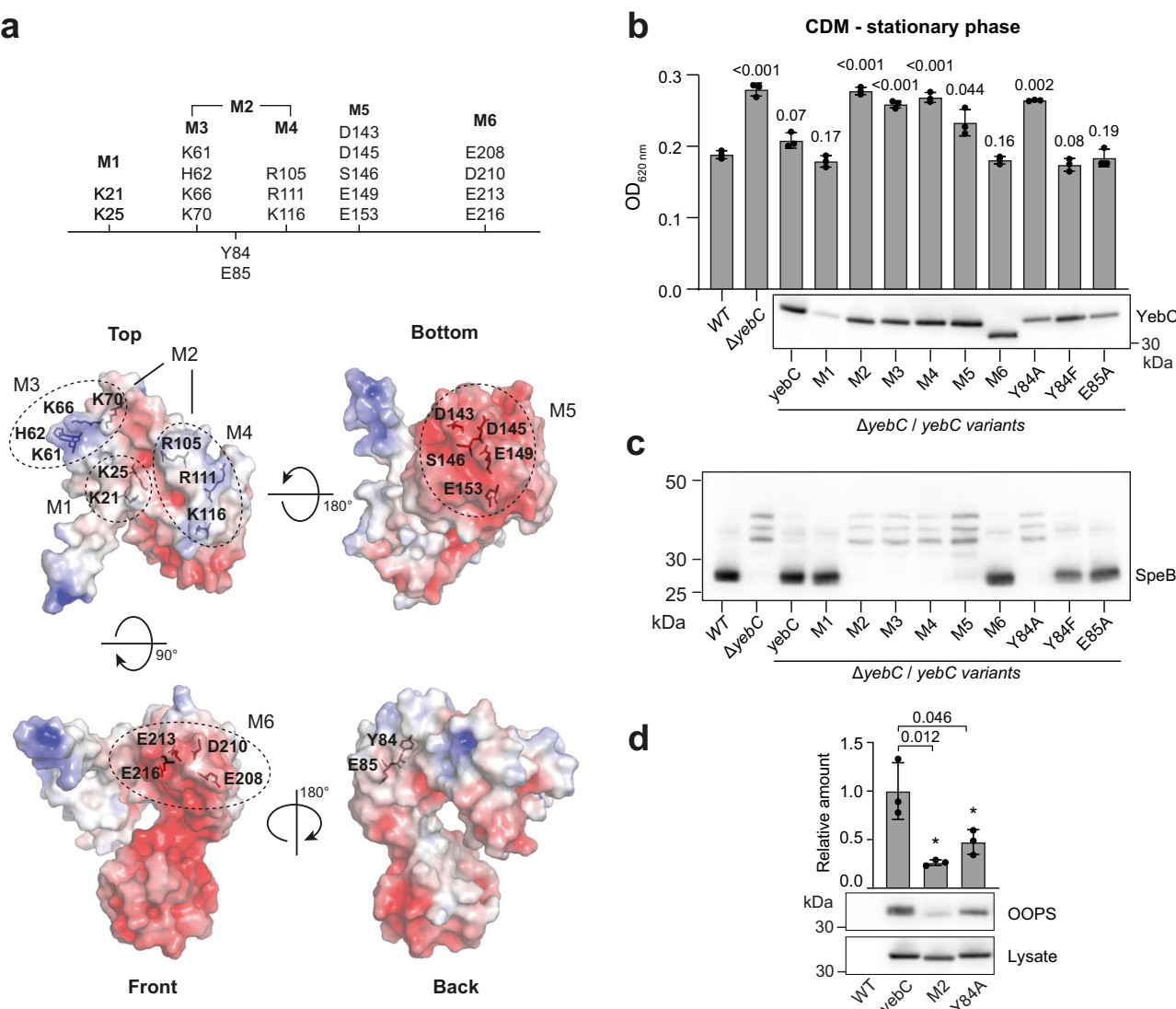

**Fig. 4 | Identification of amino acid residues critical for YebC acivity. a** Amino acid residues for site-directed mutagenesis superimposed on the YebC structure. **b** Optical density of stationary phase cultures of *yebC* mutant and complemented strains in CDM. Data represent mean ± SD of three biological replicates. The optical density of cultures of each strain was compared to that of the WT. The statistical significance of the differences was estimated using the two-sided *t* test and the *p* values are indicated above the bars. The complemented YebC is tagged with the C-terminal 3x FLAG tag and its expression was measured by western blotting using anti-FLAG antibodies (shown below). **c** Expression of the SpeB protein in the *yebC* mutant and complemented strains. The strains were grown overnight in THY, the supernatants were collected and SpeB expression was probed with anti-SpeB

antibodies. The experiment was performed in two biological replicates with similar results. **d** The relative amounts of YebC and its mutants, M2 and Y84A, in the OOPS protein fraction enriched with RBPs. The Δ*yebC* strains complemented with either *yebC::3xFLAG* or its mutant versions were grown in CDM until the mid-logarithmic phase and UV irradiated. OOPS was performed and the amount of YebC in the lysate and the fraction enriched with RBPs was probed by western blotting using anti-FLAG antibodies. The relative amounts of YebC versions were measured by densitometry and normalised to the average amount of the wild-type. Data represent the mean ± SD of three biological replicates. The statistical significance of difference in amounts of the WT and mutant YebC versions was estimated using the two-sided *t* test and the *p* values are indicated above the bars.

translation of the P5 version resulted in a product of higher mass. After treatment with RNase A, the mass of the P5 product decreased to approximately 7.1 kDa, which is the mass of FolA with a premature stop codon in place of the P5 sequence. These data indicate that ribosomes almost completely stall at the pentaproline amino acid stretch and cannot synthesise the full-length FolA. The addition of YeeN to the reaction partially rescued the ribosome stalling and enabled the full-length protein to be synthesised, albeit with a much lower efficiency. Importantly, the M2 and Y84A mutants were completely inactive, demonstrating the specificity of YeeN activity in our system. Interestingly, YebC showed higher activity than YeeN: the stalled product was not visible and the amount of the full-length protein was higher. The observed difference in activity between YebC and YeeN suggests that these proteins are functionally divergent. We also investigated, whether YeeN stably associates with the ribosome in the PURE system, but no stable association was detected (Supplementary Fig. 9b). In conclusion, our experiments with in vivo and in vitro reporters demonstrate the ability of YebC to alleviate ribosome stalling at polyproline amino acid stretches.

To investigate a possible redundancy between YebC and EF-P, we wanted to examine the phenotype of the double *efp yebC* deletion. For unclear reasons, we were unable to delete *efp* in *S. pyogenes*. Therefore, we opted for *Salmonella* Typhimurium as a model organism, in which the *efp* deletion only slightly reduces the growth rate. The genome of *S.* Typhimurium encodes two *yebC* paralogs: *yebC* (*yebC_I*) and *yeeN* (*yebC_II*) (Fig. 2d). Deletion of *yeeN* in both WT and Δ*efp* backgrounds and deletion of *yebC* in the WT background did not affect the growth rate (Fig. 8b and Supplementary Fig. 13). Interestingly, we were unable

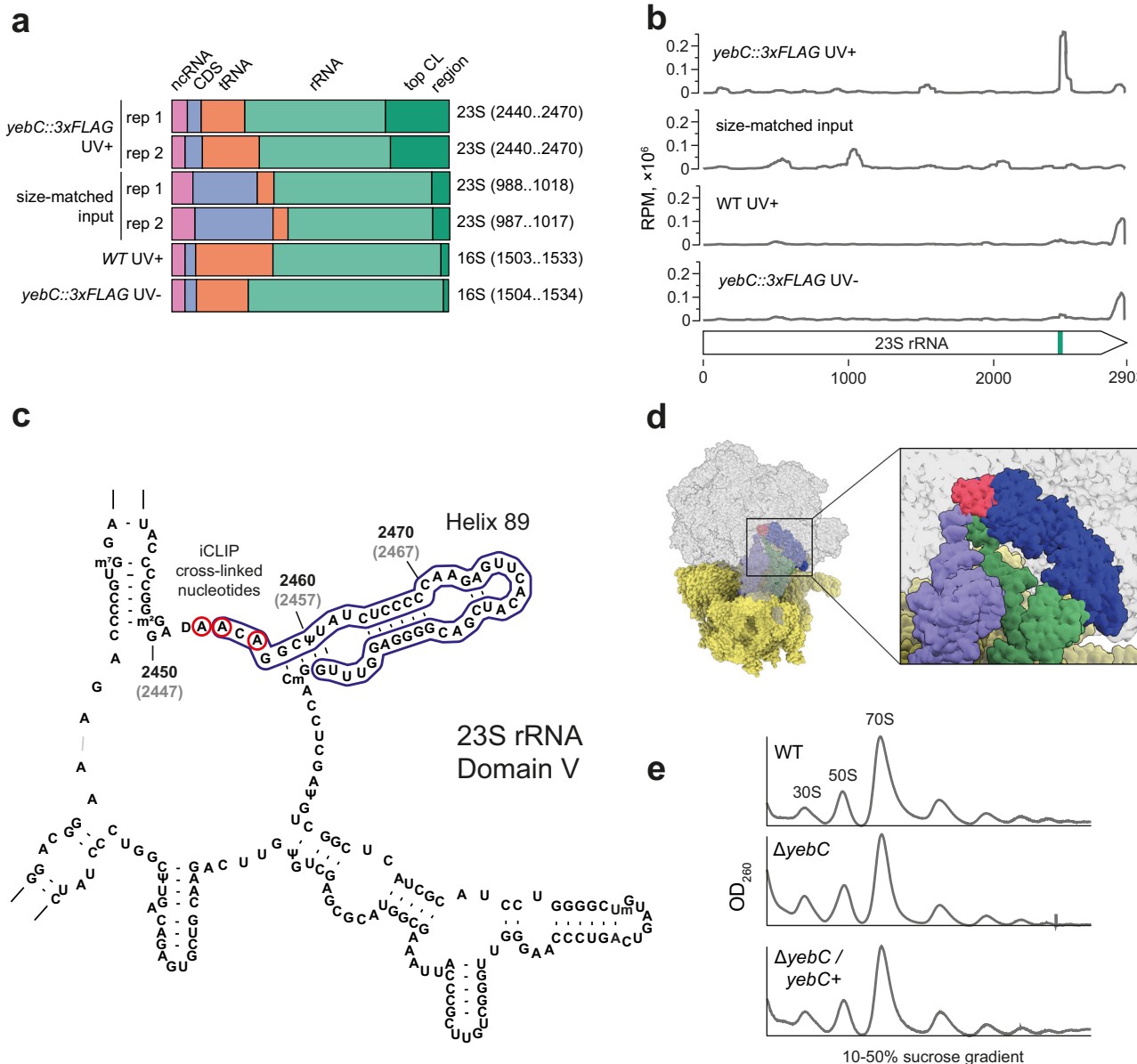

**Fig. 5 | Interaction of YebC with the ribosome. a** Distribution of the nucleotides cross-linked to YebC in the *S. pyogenes* transcriptome. Genomic regions (30 nt in length) showing an increased number of cross-linked nucleotides relative to neighbouring regions were determined. The region with the highest number of cross-linked nucleotides (top CL region) for each library is highlighted on the right. **b** iCLIP read coverage of 23S rRNA in *yebC::3xFLAG* UV+ and control samples. Coverage was normalised to the sequencing depth of the libraries and presented as RPM values. The top cross-linking region in *yebC::3xFLAG* UV+ encompassing nucleotides 2440 to 2470 is marked in green on the 23S rRNA. **c** Position of the cross-linking region in domain V of 23S rRNA. The nucleotides with increased iCLIP read coverage are circled in blue and the cross-linked nucleotides are circled in red. The coordinates of nucleotides in *S. pyogenes* are shown in black and the coordinates of the homologous nucleotides in *E. coli* are shown in grey. **d** Structure of *E. coli* ribosome (PDB ID 7K00) with transparent (left) and transverse (right) section of the 50S (grey) to reveal density for helix 89 (blue) with A2450, A2451 and A2452 highlighted in red, P-tRNA (lavender), A-tRNA (green) and 30S (yellow). **e** Sucrose density gradient ribosome traces of WT, Δ*yebC* and Δ*yebC/yebC+* strains.

to delete *yebC* in the Δ*efp* background. Therefore, we used CRISPR interference (CRISPRi) to deplete *yebC* in the WT and Δ*efp* strains[50,51]. Due to leaky expression of sgRNA and dCas9 in our experimental setting, *yebC* was depleted even without the addition of inducer (Supplementary Fig. 14). As expected, *yebC* depletion in the WT background had no effect on the bacterial growth rate (Fig. 8c). However, in the Δ*efp* background, *yebC* depletion significantly decreased the growth rate. These results suggest that EF-P and YebC (YebC_I) are functionally redundant, whereas YeeN (YebC_II) is not redundant with either of them. This corroborates our in vitro translation experiment, where YebC was more efficient than YeeN in resolving ribosome stalling at the polyproline motif.

## Discussion

In this study, we used the OOPS and RBS-ID approaches to identify previously uncharacterised RBPs in *S. pyogenes*. A combination of these techniques reliably identified the majority of annotated RBPs, as well as a group of proteins that had not previously been shown to interact with RNA. Our list of RBP candidates in *S. pyogenes* includes 30 proteins (Table 1). Several of these proteins have previously been classified as novel RBPs in bacteria[3–8,34–36]. Ten of the identified RBP candidates are of unknown function. We selected five of these and tested their cross-linking with RNA using immunoprecipitation and radioactive labelling of co-immunoprecipitated RNA. The protein PhoH showed weak cross-linking to RNA. The PhoH domain belongs to

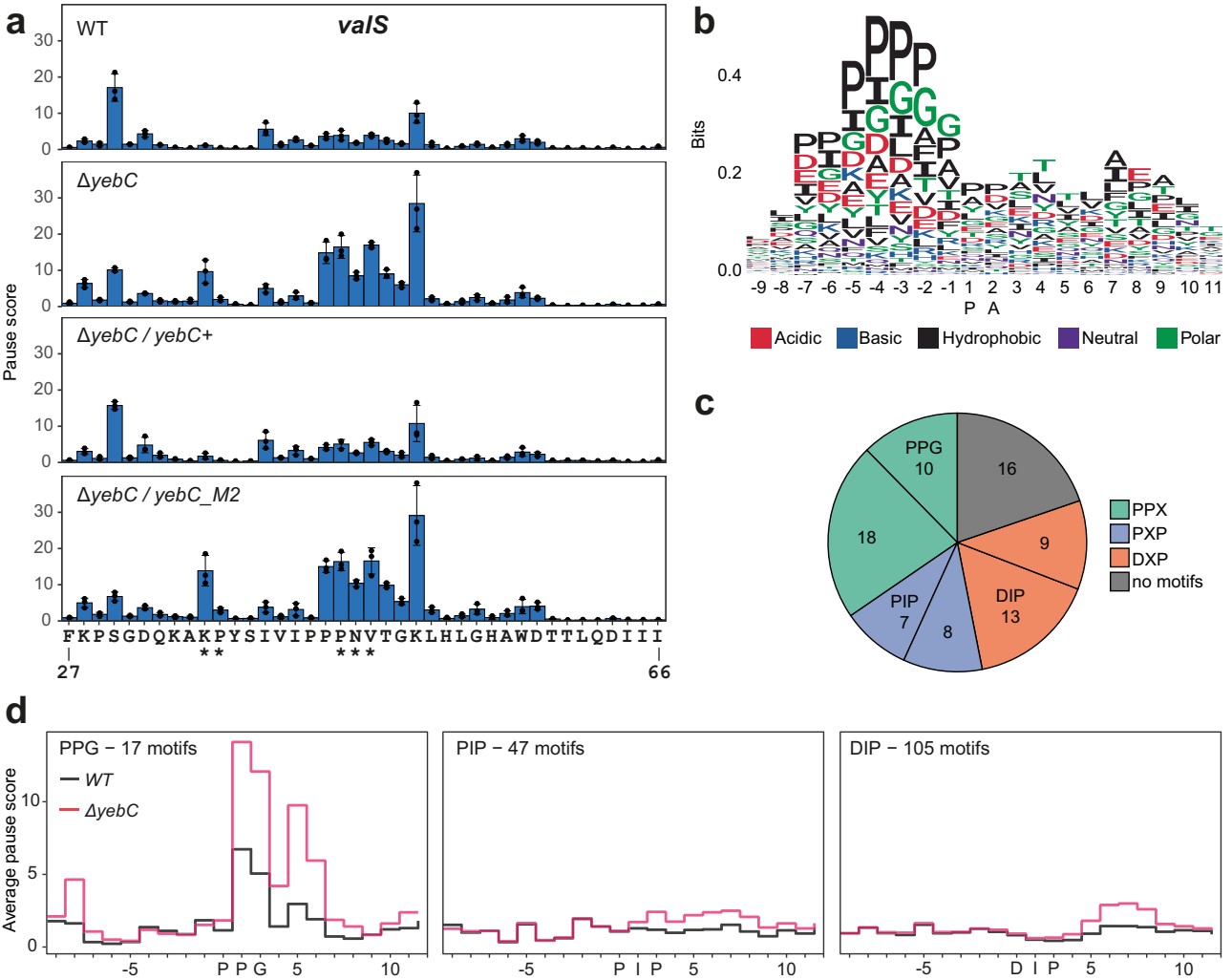

**Fig. 6 | Ribosome pausing in *S. pyogenes yebC* mutants. a** Mean pause scores for *valS* codons 27-66 in the WT and *yebC* mutant strains. Data represent mean ± SD of three biological replicates. Codons with a statistically significant difference in pause scores between the strains with functional and non-functional YebC are indicated by asterisks. The criteria for estimation of statistical significance are described in Methods. **b** Logo plot of codons in the vicinity of sites with increased pausing in the *yebC* mutant strains. The x-axis shows the position of the paused ribosome with the P and A sites indicated. **c** Motifs enriched in the vicinity of sites with increased pausing in the *yebC* mutant strains. **d** Average pausing scores for the codons surrounding PPG, PIP and DIP motifs. The occurrence of these motifs in *S. pyogenes* mRNAs is indicated.

the P-loop NTPases and has been proposed to function as an RNA helicase[52]. Two other RBP candidates, YgaC and ThuC, also showed cross-linking to RNA, but were expressed at low levels under the condition tested. YgaC contains the DUF402 domain and its homologue in *S. aureus* possesses nucleoside diphosphatase activity[53]. ThuC is one of the proteins encoded by a biosynthetic gene cluster that is involved in the synthesis of a secondary metabolite[54,55]. Two other RBP candidates, YjbK and YebC, were highly expressed and exhibited robust cross-linking with RNA. YjbK belongs to the CYTH domain superfamily – the metal-dependent phosphohydrolases, whose active site is located within a topologically closed hydrophilic beta-barrel. The representatives of this superfamily have diverse activities. In fungi and protozoa, it is an RNA triphosphatase that catalyses the first step of RNA capping; in mammals, it is a thiamine triphosphatase; and in *E. coli*, a CYTH-domain protein demonstrates triphosphatase activity[56–58].

Among the RBP candidates, we selected YebC for further characterisation. YebC is an evolutionarily conserved protein with very similar structures in species as evolutionarily distantly related as the thermophilic bacterium *Aquifex aeolicus* and the mouse[13]. We show that the amino acid residues forming positively charged surfaces of domains I and II are essential for YebC activity. These surfaces are

located in close proximity to the identified RNA cross-linking sites and could be involved in the interaction with RNA. We also tested the invariant Y84 and E85 positions. Surprisingly, mutation of E85 did not affect the ability of the protein to complement the Δ*yebC* phenotype in *S. pyogenes*. For Y84, substitution by an alanine, but not a phenylalanine, rendered YebC unable to complement the Δ*yebC* phenotype. Given that Y84 represents an RNA cross-linking site, we hypothesise that the aromatic ring of the tyrosine residue might be important for interaction with RNA.

Our data suggest that YebC interacts with the ribosome. According to the iCLIP results, the protein cross-links with the domain V of 23S rRNA. iCLIP relies on reverse transcriptase stopping at the cross-linking site between the RNA and peptides remaining after protein digestion. Consequently, YebC is likely to cross-link with the nucleotides constituting the PTC: A2453, A2454 and A2456. However, in some cases reverse transcriptase does not terminate synthesis at the cross-linked nucleotides[59], and we cannot exclude the possibility that YebC cross-links with a nucleotide located in helix 89 of 23S rRNA. Despite the robust cross-linking, YebC does not appear to form a stable complex with the ribosome, suggesting that their interaction is rather transient. Some translation factors also interact with the

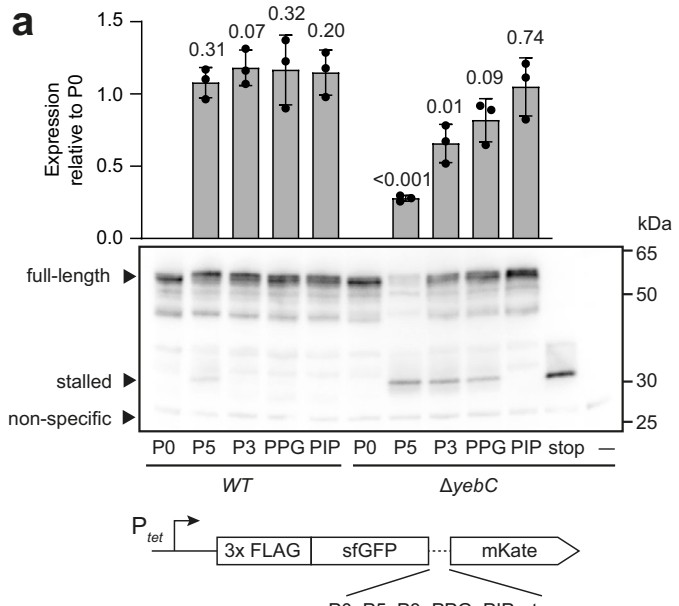

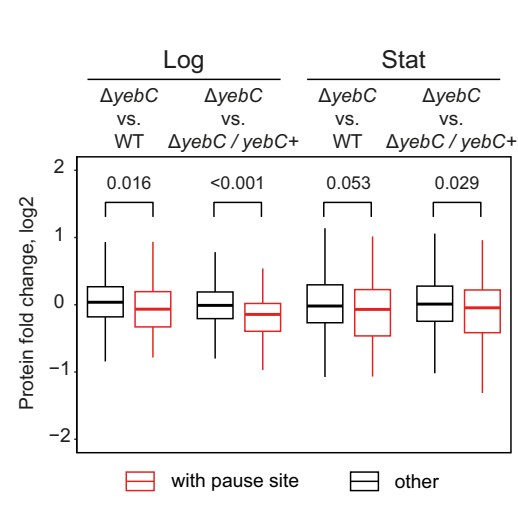

**Fig. 7 | Effect of YebC on translation of proline-rich regions in *S. pyogenes*.**
**a** Effect of *yebC* deletion on the polyproline stalling in vivo. The reporter proteins bearing different amino acid sequences between sfGFP and mKate were introduced to the WT and Δ*yebC* strains under the control of the P*tet* promoter. Expression of the reporter was induced in the exponentially growing *S. pyogenes* cells, and the cells were collected 30 min after induction. Western blotting with anti-FLAG antibodies detected the full-length reporter, the truncated reporter with stop codon and the truncated reporters in the Δ*yebC* strain. The non-specific band served as a loading control. The amount of the full-length products relative to P0 was measured by densitometry and presented in the barplot as mean ± SD of three biological replicates. The statistical significance of the difference relative to the P0 strain

was estimated with the two-sided *t* test. **b** Genes with YebC-dependent pausing are expressed at a lower level in the Δ*yebC* strain. The protein expression in the mid-logarithmic (Log) and stationary (Stat) growth phases was measured by MS proteomics in three biological replicates. The average expression level of each protein in the Δ*yebC* strain was compared with that in the WT and Δ*yebC*/*yebC*+ strains. The fold change differences for proteins with YebC-dependent pause sites (*n* = 69) and without them (*n* = 1246) are presented as a boxplot, where the box represents the interquartile range and the median, and the whiskers extend to ×1.5 of the interquartile range. The statistical significance of the difference in protein expression was estimated with the two-sided Mann-Whitney test.

ribosome transiently, and isolation of their interaction intermediates may require the mutant version of a protein or the addition of antibiotics[60,61].

The discovery of the interaction with the ribosome prompted us to examine the effect of *yebC* deletion on translation. Ribosome profiling revealed increased pausing at proline-rich motifs in the *yebC* mutant. This is similar to the pausing induced by *efp* deletion[23,47]. However, in the case of YebC, pausing was often induced by the PIP and DIP motifs, which is not typical for *efp* mutants. In addition, the ribosome often pauses a few codons after translating the polyproline motif in the *yebC* mutant. This suggests that YebC and EF-P have different mechanisms of action. Experiments with in vivo reporters in *S. pyogenes* have shown that the rescue of ribosome stalling by YebC is necessary for efficient translation. The stalling was induced by the consecutive prolines but not by the PIP motif. According to the ribosome profiling data, only some PIP motifs induce pausing and this pausing is generally weaker than that at PPG motifs. Therefore, detectable stalling at the PIP motif may only occur in a specific context.

The genes with increased ribosome pausing in the Δ*yebC* strain tend to be expressed at slightly lower levels. In general, the deletion of *yebC* had a pleiotropic effect on the physiology of *S. pyogenes*. The most obvious phenotypes were slightly altered growth and reduced expression of the virulence factor SpeB. Our results demonstrated that translation and secretion of SpeB are not affected by the deletion of *yebC*. On the other hand, the decrease in activity of the *speB* promoter seems to be the reason for down-regulation of *speB* in the Δ*yebC* strain. The *speB* promoter is regulated by a quorum-sensing pathway consisting of a secreted leaderless peptide signal, and its cognate receptor RopB[62]. Acidification of the environment increases the affinity of RopB for the leaderless peptide and serves as a second factor in increasing

the expression of SpeB[63]. We did not identify a YebC-dependent pausing in *speB* or *ropB*, and the observed deficiency in SpeB expression is probably an indirect effect of the *yebC* deletion. Consistent with SpeB playing a role in the intracellular persistence[64] and cleavage of pro-IL-1β[65], the strain with deleted *yebC* showed reduced survival in human macrophages and elicited a slight decrease in the release of IL-1β and IL-18.

It has been suggested that the YebC proteins are represented by two subtypes: YebC_I and YebC_II[39]. While the genome of *S. pyogenes* encodes only YebC_II, some other bacterial species, for example *E. coli* and *S.* Typhimurium possess both YebC subtypes. We studied the activity of the YebC_I and YebC_II proteins of *E. coli* using an in vitro translation system. Both paralogs were able to resolve the ribosome stalling at the polyproline motif, but YebC_I was more efficient. We also tested the functional redundancy of EF-P and YebC paralogs in *S.* Typhimurium. While deletion of *yebC_II* in the Δ*efp* background did not affect bacterial growth, depletion of *yebC_I* significantly decreased the growth rate of the Δ*efp* strain. These results highlight the difference in activity between YebC_I and YebC_II and suggest that a certain degree of functional diversification exists between the YebC subtypes.

In general, our results demonstrate that the activity of YebC is similar to the activity of its mitochondrial ortholog, TACO1[15]. In a recent publication, YebC2 (YeeI) has also been shown to alleviate stalling at the polyproline motifs in *Bacillus subtilis*[66]. The molecular mechanism of action for YebC is not clear and its elucidation may require structural studies. Our data suggest that in bacteria, YebC interacts with the ribosome in a transient manner and the isolation of their complex for structural studies may therefore be a non-trivial task. However, it is likely that YebC interacts with the PTC: in *S. pyogenes*, the protein cross-links with the PTC nucleotides, and in mitochondria,

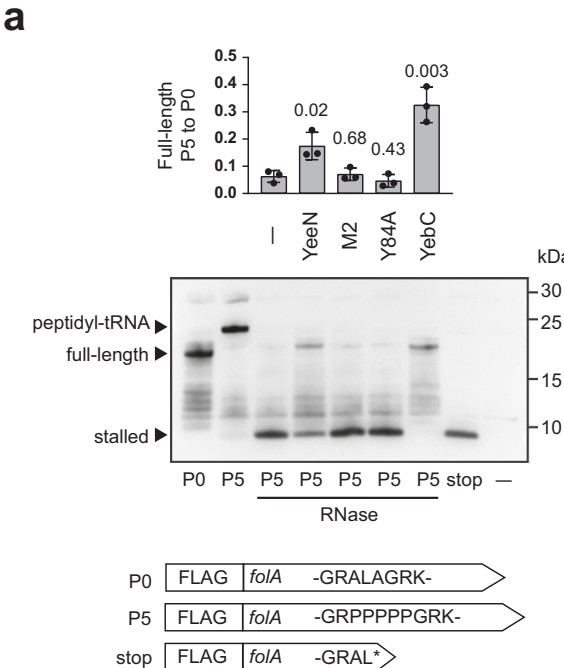

**Fig. 8 | Activity of YebC paralogs. a** Effect of YebC paralogs on the polyproline stalling in vitro. The *folA* mRNA with the indicated mutations served as a template for in vitro translation using the PURE system. The *E. coli* proteins YeeN, its mutants and YebC were added to the reactions with P5 mRNA. The indicated reactions were then treated with RNase A. The barplot represents the amount of the full-length product in P5, as measured by densitometry and normalised to P0, with values representing the mean ± SD of three independent replicates. A significance of the effect of YebC proteins on the amount of the full-length product was estimated using the two-sided *t* test. **b** The growth rates of *S*. Typhimurium mutants. The

calculated growth rates are presented in the barplot as mean ± SD of six biological replicates. **c** The growth rates of *S*. Typhimurium mutants upon *yebC* depletion using CRISPRi. *S*. Typhimurium WT and Δ*efp* strains were transformed with either a vector expressing dCas9 or a vector expressing both dCas9 and a sgRNA targeting *yebC*. The cells were grown in LB supplemented with 10 μg/mL chloramphenicol with or without inducers of dCas9 and sgRNA (10 mM arabinose and 100 ng/mL anhydrotetracycline). The calculated growth rates are presented in the barplot as mean ± SD of six biological replicates. The statistical significance of the differences in growth rates was estimated with the two-sided *t* test.

TACO1 has been proposed to stabilise the PTC during translation together with the ribosomal protein bL27m[15]. It is remarkable that in bacteria, the ribosome requires the action of three specialised proteins – EF-P, YfmR and YebC−for efficient translation of proline-rich motifs. Future studies should address how these proteins act in a coordinated manner.

## Methods

### *S. pyogenes* culture and mutagenesis

*Streptococcus pyogenes* strain SF370; M1 GAS and its derivatives were cultured on Trypticase soy agar (BD Difco) plates supplemented with 5% defibrinated sheep blood. In THY, C medium and the chemically defined medium (CDM)[29], the bacteria were cultured statically at 37 °C and 5% $CO_2$. The amount of SpeB in the culture supernatant was probed by western blotting with anti-SpeB antibodies (Thermo Scientific; PA5-117551). The microscope images were acquired using Leica DMi8 microscope with the 100X phase contrast objective (HC LP APO 100×/1.40 oil) and 50 ms exposure time. Cell area was calculated using MicrobeJ[67].

The introduction of 3x FLAG to the C-termini of *S. pyogenes* genes was performed using the Cre-Lox recombination system[68]. The *S. pyogenes* suicidal plasmid pSEVA141_3x FLAG-lox71-*erm*-lox66 (Supplementary Fig. 15) encodes the 3x FLAG tag sequence and the lox71-*erm*-lox66 cassette. The 3x FLAG sequence is N-terminally flanked by the KpnI restriction site allowing in-frame cloning of coding sequences and terminates with a stop codon. For each gene, the sequences upstream and downstream of their stop codons were amplified from *S. pyogenes* genomic DNA using primers listed in Supplementary Table 1. The primers introduced the cutting sites for restriction enzymes. To generate the plasmids for the 3x FLAG introduction (Supplementary

Table 2), the upstream fragments were cloned in-frame with the 3x FLAG, and the downstream fragments were cloned downstream of the lox71-*erm*-lox66 cassette. The competent *S. pyogenes* cells for transformation with the suicidal vectors were prepared. For that, the cells were grown in THY supplemented with 250 mM sucrose and 40 mM L-threonine (Sigma-Aldrich) until the mid-logarithmic phase. The cells were washed with 0.5 M sucrose and resuspended in 0.5 M sucrose and 20% glycerol. 15 μg of plasmids for 3x FLAG introduction were digested with Cfr42I and introduced by electroporation in a 0.1 cm electrode gap cuvette with a 1.8 kV, 400 Ω and 25 μF pulse. After 2 h of growth in THY, the cells were plated on TSA plates with 2.5 μg/mL erythromycin. After two days, the individual colonies were re-streaked to the fresh erythromycin plates and incubated for an additional day. The re-grown colonies were inoculated into fresh THY and grown over day, the clones were collected and the glycerol stocks were prepared. For excision of the lox71-*erm*-lox66 cassette, the clones were transformed with the replicative pEC455 plasmid (Supplementary Table 2) encoding the Cre recombinase. For that, the cells were grown in THY until the mid-logarithmic growth phase. The cells were washed with ice-cold mQ, resuspended in 20% glycerol and electroporated with 0.5 μg of the pEC455 plasmid. After 2 h of growth in THY, the cells were plated on TSA plates with 250 μg/mL kanamycin. After two days, the individual colonies were inoculated into THY and grown over day to lose the pEC455 plasmid. The cultures were streaked on TSA plates without antibiotics and allowed to grow overnight. The colonies were inoculated into fresh THY and grown over day, the clones were collected and the glycerol stocks were prepared (Supplementary Table 3). The loss of the *erm* resistance cassette and pEC455 was confirmed by streaking the cultures on erythromycin and kanamycin plates. The introduction of C-terminal 3x FLAG followed by the lox72 site was confirmed by

isolation of genomic DNA with NucleoSpin Microbial DNA kit (Macherey-Nagel), amplification of the corresponding genomic regions with primers listed in Supplementary Table 1 and Sanger sequencing of the PCR fragments with the same primers.

The deletion of the *yebC* and *speB* genes was also carried out using the Cre-Lox recombination system. The *S. pyogenes* suicidal plasmid pSEVA141_lox71-*erm*-lox66 (Supplementary Fig. 15) bears the lox71-*erm*-lox66 cassette. The sequences upstream and downstream of the genes were amplified from *S. pyogenes* genomic DNA using primers listed in Supplementary Table 1. The primers introduced the cutting sites for restriction enzymes. Using the restriction enzymes, the fragments were cloned upstream and downstream of the lox71-*erm*-lox66 cassette to generate the plasmids for *yebC* and *speB* deletion (Supplementary Table 2). The substitution of the *yebC* and *speB* genes with the lox71-*erm*-lox66 cassette and excision of the cassette were performed according to the same protocol as for the introduction of 3x FLAG to the C-termini of the genes.

The WT and mutant versions of *yebC* were introduced into the *S. pyogenes* Δ*yebC* strain with the modified p7INT integrative vector[69]. The p7INT portion without the *lacZ* promotor and coding sequence, and *yebC* locus including the promotor and terminator were amplified with the primers listed in Supplementary Table 1. The primers introduced the cutting sites for restriction enzymes that were used for cloning. Next, the 3x FLAG sequence was introduced to the C-terminus of *yebC* by the whole-plasmid PCR and circularisation with KLD Enzyme Mix (New England Biolabs). The mutations to the *yebC* sequence were also introduced by the whole-plasmid PCR with respective primers followed by circularisation with KLD Enzyme Mix. The M2 mutation is a combination of M3 and M4 and was created by two rounds of PCR mutagenesis. The resulting p7INT derivatives (Supplementary Table 2) were introduced into the *S. pyogenes* genome by electroporation and antibiotic selection. The competent cells of the Δ*yebC* strain were prepared in the same way as for the transformation with the pEC455 plasmid. The cells were transformed with 0.5 μg of p7INT derivatives. After 2 h of growth in 5 mL of THY, the cells were plated on TSA plates with 2.5 μg/mL erythromycin. The colonies were re-streaked onto the fresh erythromycin plates and incubated for an additional day. The re-grown colonies were inoculated into fresh THY and grown over day, the clones were collected and the glycerol stocks were prepared.

To study *speB* regulation, the reporter plasmids were assembled and introduced into the *WT*, Δ*yebC*, Δ*speB* and Δ*yebC* Δ*speB* strains. The plasmid p7INTΔlacZα_pTet_sfGFP encodes the sfGFP protein under the control of the pTet promoter in the p7INT backbone (Supplementary Fig. 16). This plasmid was transformed into the WT and Δ*yebC* strains using the same protocol as for the complementation with the *yebC* protein. The *speB* coding sequence was amplified from *S. pyogenes* genomic DNA with primers HiFi_SpeB_F and R (Supplementary Table 1) and cloned into the p7INTΔlacZα_pTet_sfGFP plasmid amplified with primers HiFi_pTet_F and R using HiFi DNA assembly mix (Thermo Scientific). The resulting plasmid p7INTΔlacZα_pTet_speB was transformed into the Δ*speB* and Δ*yebC* Δ*speB* strains. The sequence of *speB* promoter was amplified from *S. pyogenes* genomic DNA with primers HiFi_pSpeB_F and R and cloned into the p7INTΔlacZα_pTet_sfGFP plasmid amplified with primers HiFi_GFP_F and R. The resulting plasmid p7INTΔlacZα_pSpeB_sfGFP was introduced into the WT and Δ*yebC* strains.

The reporter plasmids for ribosome stalling were assembled and introduced into *S. pyogenes* genome. The plasmid p7INTΔlacZα_pTet_3x FLAG-sfGFP-P0-mKate encodes the fusion protein 3x FLAG-sfGFP-mKate2 under the control of the pTet promoter in the p7INT backbone (Supplementary Fig. 16). The sfGFP and mKate amino acid sequences are connected via a Gly-Ser (P0) linker. To change the linker sequence, the whole-plasmid PCR was performed with the following primers: (P5) P5_F and P5_P3_R; (P3) link_F and P5_P3_R; (PPG) link_F and PPG_R; (PIP) link_F and PIP_R; (stop) link_F and stop_R (Supplementary

Table 1). The products were circularised with the KLD enzyme mix (New England Biolabs). The resulting vectors (Supplementary Table 2) were transformed into the WT and Δ*yebC* strains.

## S. Typhimurium culture and mutagenesis

*Salmonella enterica* serovar Typhimurium strain LT2 and its derivatives were cultured on lysogeny broth (LB) agar plates at 37 °C. In LB liquid medium, the bacteria were cultured at 37 °C and under constant shaking (180 rpm).

Scarless gene deletions were performed using a two-step λ-Red recombineering system[70]. The method involves the insertion of a Kan-I-SceI selection-counterselection cassette at the target locus, followed by its replacement with a clean deletion. The λ-Red recombination plasmid pWRG730 was electroporated into *S.* Typhimurium, and transformants were selected on LB agar supplemented with 10 μg/mL chloramphenicol at 30 °C. The plasmid was maintained throughout the mutagenesis procedure. For recombination, electrocompetent cells were prepared by growing *S.* Typhimurium carrying pWRG730 in LB with chloramphenicol (10 μg/mL) at 30 °C to an optical density at 600 nm of 0.4–0.5. Cultures were then subjected to a heat-shock at 42 °C for 12 min 30 s with shaking (180 rpm) to induce expression of the λ-Red genes, followed by incubation on ice for 15 min. Cells were harvested by centrifugation (10,000 × *g*, 5 min, 4 °C), washed twice with ice-cold ultrapure water, and resuspended in water for electroporation. The Kan-I-SceI cassette was PCR-amplified using primers with 40-bp homology arms flanking the target locus (Supplementary Table 1). The purified PCR product was electroporated into the prepared cells, which were recovered in LB at 30 °C for 60 min before plating on LB agar with kanamycin (25 μg/mL) and incubating overnight at 30 °C. The next day, the transformants were re-streaked on LB with chloramphenicol and again incubated overnight at 30 °C. The correct integration of the cassette was confirmed by colony PCR. To verify the I-SceI functionality, the positive clones were streaked in parallel on LB with chloramphenicol as well as on LB with chloramphenicol and anhydrotetracycline (100 ng/mL), followed by incubation overnight at 30 °C. Clones exhibiting a growth defect on anhydrotetracycline-containing plates were selected for further processing. Afterwards, a PCR fragment corresponding to the clean deletion allele was amplified using overlapping primers with 40-bp homology arms flanking the target locus and electroporated into *S.* Typhimurium strains carrying both pWRG730 and the Kan-I-SceI cassette. Following the transformation, the cells were recovered by shaking at 30 °C for 45 min. Anhydrotetracycline (100 ng/mL) and chloramphenicol (10 μg/mL) were added to induce I-SceI expression from a *tet*-promoter and maintain pWRG730, and incubation continued for another 45 min before plating serial dilutions on LB with chloramphenicol and anhydrotetracycline and incubation at 30 °C overnight. The following day, clones exhibiting optimal growth on anhydrotetracycline-containing plates were re-streaked onto fresh LB with chloramphenicol and anhydrotetracycline and again incubated overnight at 30 °C. For curing of pWRG730, the clones were streaked onto LB without antibiotics and inducers and incubated overnight at 42 °C. Loss of pWRG730 was verified by screening for chloramphenicol sensitivity. Final confirmation of scarless deletion was performed by colony PCR and sequencing of the kanamycin sensitive clones.

To inhibit *yebC* expression in the WT and Δ*efp* strains, the CRISPR interference was used. The oligonucleotides gRNA-yebC-nts_F and R were annealed and cloned into the plasmid pdCas9-sgRNA-RFP[51] digested with BsaI (Thermo Scientific). The resulting plasmid pdCas9-sgRNA-yebC and the original plasmid were introduced into *S.* Typhimurium strains by electroporation and selection with 10 μg/mL chloramphenicol. The expression of the sgRNA from the arabinose-promoter and the dCas9 from the *tet*-promoter was induced by supplementing the growth medium with 0.2% arabinose and 100 ng/mL anhydrotetracycline. To determine the bacterial growth of *S.*

Typhimurium mutants with and without CRISPRi (Supplementary Table 4), pre-cultures were grown to stationary phase, diluted 1:100 in either LB or selective medium with and without supplements, and the optical density at 600 nm was measured using a microplate reader (Tecan Infinite M200).

## Orthogonal Organic Phase Separation (OOPS)

*S. pyogenes* cells were grown in CDM until the mid-logarithmic phase. Each culture was grown in two batches: in one batch, CDM was supplemented with $^{13}C^{15}N$-labelled L-lysine and L-arginine; in the other batch—with the non-labelled amino acids. The cells were rapidly cooled in an ice-water bath and collected by centrifugation. The cells were re-suspended in the ice-cold PBS and transferred to the cooled Petri dishes. One batch was UV irradiated with 600 mJ cm$^{-2}$ UV 254 nm light on the Bio-Link BLX cross-linker (Vilber), and the other batch was kept on ice. The UV+ and UV− batches were merged and the cells were collected by centrifugation for 5 min at $4500 \times g$ +4 °C. The experiment was performed in four replicates: in two of the replicates, the UV+ culture was SILAC labelled, and in the other two replicates, the UV− culture was labelled.

The cells were disrupted with 0.1 mm glass beads (Roth) on Fastprep-24 5G (MP Biomedicals). Aliquots of the lysates were collected and analysed by mass-spectrometry (Input samples). The rest of the lysates were subjected to acid guanidinium thiocyanate-phenol-chloroform extraction, and the interfaces between the organic and the aqueous phases were collected as described in Queiroz et al.[4]. Briefly, the interfaces were subjected to two more rounds of acid guanidinium thiocyanate-phenol-chloroform extraction. The third interfaces were mixed with methanol and precipitated by centrifugation for 10 min at $14,000 \times g$. The pellets were washed with methanol, dried on the bench and dissolved in 100 μL of RBP buffer (100 mM TEAB, 1 mM MgCl$_2$, 1% SDS). The samples were incubated at 95 °C for 20 min and cooled to room temperature. RNA was degraded overnight using the RNase A/T1 mix (Thermo Scientific), and the samples were subjected to acid guanidinium thiocyanate-phenol-chloroform extraction. The organic phases were collected, mixed with 9 vol. of methanol and the proteins were precipitated by centrifugation. The proteins were solubilized in 1x Laemmli sample buffer without dye by incubation at 95 °C for 10 min.

The samples were analysed on an Orbitrap Fusion Lumos (Thermo Scientific) that was coupled to a 3000 RSLCnano UPLC (Thermo Scientific). Samples were loaded on a PepMap Trap cartridge (300 μm i.d. x 5 mm, C18, Thermo Scientific) with 2% acetonitrile, 0.1% TFA at a flow rate of 20 μL/min. Peptides were separated over a 50 cm analytical column (PicoFrit, 360 μm O.D., 75 μm I.D., 10 μm tip opening, non-coated, New Objective) that was packed in-house with Poroshell 120 EC-C18, 2.7 μm (Agilent). Solvent A consists of 0.1% formic acid in water. Elution was carried out at a constant flow rate of 250 nL/min using a 180-min method: 8–33% solvent B (0.1% formic acid in 80% acetonitrile) within 120 min, 33–48% solvent B within 25 min, 48–98% buffer B within 1 min, followed by column washing and equilibration. The Orbitrap Fusion Lumos mass spectrometer was equipped with a FAIMS Pro device, which was operated at standard resolution using three alternating CVs of −40 V, −60 V and −80 V (cycle time for each was set to 1 s). Data acquisition was carried out using data-dependent acquisition in the positive ion mode. MS survey scans were acquired from 375–1500 m/z in profile mode at a resolution of 240,000. AGC target was set to 4e$^5$ charges, allowing a maximum injection time of 50 ms. Peptides with charge states 2–6 were subjected to CID fragmentation (fixed CE = 35%, AGC = 1e$^4$) and analysed in the linear ion trap at a resolution of 125,000 Da/second. The isolation window was set to 1.6 m/z.

The calculated SILAC ratios were log2 transformed. In order to correct for mixing errors, the median SILAC ratio of the Input samples was determined and subtracted from the ratio of the corresponding OOPS samples. Proteins with a SILAC ratio count less than 2 were

removed. Next, quantile normalisation was performed separately for each experiment and the resulting OOPS enrichment values were used for further analysis. Statistical significance of OOPS enrichment was assessed using the R package Limma[71]. Resulting *p*-values were corrected for multiple testing employing the approach by Benjamini-Hochberg[72]. Proteins with an adjusted $p < 0.05$ were considered to have statistically significant OOPS enrichment.

## RBS-ID

*S. pyogenes* cells were grown in CDM with non-labelled L-Lysine and L-arginine until the mid-logarithmic phase and UV irradiated the same way as the OOPS samples. The cells were disrupted and three rounds of acid guanidinium thiocyanate-phenol-chloroform extraction were performed according to the OOPS protocol. After methanol precipitation, the proteins from the third interface were reduced and alkylated with 2-chloroacetamide. RBS-ID was performed according to Bae et al.[31]. The proteins were digested with Trypsin/LysC mix (Promega) and the RNA-peptide conjugates were purified using the RNeasy Midi kit (Qiagen). RNA was degraded with 4 vol. of 48% hydrofluoric acid at 4 °C overnight. The samples were dried with SpeedVac vacuum concentrator (Thermo Scientific) that was supplemented with CaCO$_3$ trap and located under the fume hood.

The samples were analysed on an Orbitrap Exploris 480 (Thermo Scientific) and Orbitrap Fusion Lumos, which were both coupled to a 3000 RSLCnano UPLC (Thermo Scientific). Samples were loaded on a PepMap Trap cartridge (300 μm i.d. × 5 mm, C18, Thermo Scientific) with 2% acetonitrile, 0.1% TFA at a flow rate of 20 μL/min. Peptides were separated over a 50 cm analytical column (PicoFrit, 360 μm O.D., 75 μm I.D., 10 μm tip opening, non-coated, New Objective) that was packed in-house with Poroshell 120 EC-C18, 2.7 μm (Agilent). Solvent A consists of 0.1% formic acid in water. Settings for Exploris: Elution was carried out at a constant flow rate of 250 nL/min within 90 min. Initially, a two-step linear gradient was applied: 3–30% solvent B (0.1% formic acid in 80% acetonitrile) within 74 min, 30–45% solvent B within 14 min, followed by column washing and equilibration. Data acquisition was carried out using data-dependent acquisition in the positive ion mode. MS survey scans were acquired from 375–1500 m/z in profile mode at a resolution of 60,000. AGC target was set to 300%, allowing a maximum injection time of 25 ms. Peptides with charge states 2–6 were subjected to HCD fragmentation (fixed CE = 30%, AGC = 200%, maximum injection time 80 ms) and analysed in the Orbitrap at a resolution of 45,000. The isolation window was set to 1.4 m/z. Settings Lumos: Elution was carried out at a constant flow rate of 250 nL/min using a 180 min method: 8–33% solvent B (0.1% formic acid in 80% acetonitrile) within 120 min, 33–48% solvent B within 25 min, 48–98% buffer B within 1 min, followed by column washing and equilibration. Data acquisition was carried out using data-dependent acquisition in the positive ion mode. MS survey scans were acquired from 375 to 1500 m/z in the profile mode at a resolution of 120,000. AGC target was set to 4e$^5$ charges, allowing a maximum injection time of 50 ms. Peptides with the charge states 2–5 were subjected to CID fragmentation (fixed CE = 35%, AGC = 2e$^5$) and analysed in the Orbitrap at a resolution of 30,000. The isolation window was set to 1.2 m/z.

The open search using MSFragger (version 3.0) within FragPipe (version 13.0)[73] resulted in identification of three major XL modifications: uridine (+244.069536 Da), uridine −H$_2$O (+226.05897 Da) and uridine −NH$_3$ (+227.042987 Da). These modifications were selected as variable modifications (considering all amino acids) in a closed search using MSFragger.

## Identification of RBP orthologs in other bacteria

To study the conservation of putative *S. pyogenes* RBPs among other bacterial species, we identified orthologous groups (OGs) and investigated the RBP content in different bacteria: *Escherichia coli*[3,4,6,7], *Salmonella typhimurium*[5], *Staphylococcus aureus*[8]. To do this, we used

the eggNOG-mapper (v2.1.12) and the eggNOG 'Firmicutes' database (v5.0.2)[74] to infer OGs. For each bacterium of interest, we downloaded the protein sequence files from RefSeq using the NCBI command-line tool 'datasets' (v16.27.2) with the following genome assembly accessions: GCF_000006785.2 (*S. pyogenes*), GCF_000005845.2 (*E. coli*), GCF_000210855.2 (*S.* Typhimurium), and GCF_000013465.1 (*S. aureus*). We used the eggNOG-mapper with default parameters to determine OGs for each protein in each species. Using the inferred OG information, we compared the *S. pyogenes* protein OGs with the OGs identified for each protein in the other bacteria. If a protein belonged to the same orthologous group and was published as a putative RBP, it was classified as 'RBP'. If a protein belonged to the same OG but was not identified as an RBP, we classified it as a homologue to the corresponding *S. pyogenes* protein but not identified as an RBP. Finally, if proteins did not belong to the same group, they were classified as nonhomologous. These analysis steps were performed using Snakemake (v8.25.3) and custom Python (v3.12) scripts.

### Immunoprecipitation and PNK labelling of *S. pyogenes* RBPs

*S. pyogenes* cells with the 3x FLAG tagged genes (Supplementary Table 3) were grown in CDM containing non-labelled L-Lysine and L-arginine until the mid-logarithmic phase. The cells were rapidly cooled in an ice-water bath, collected by centrifugation and re-suspended in the ice-cold PBS. One batch was UV irradiated with $600 \, mJ \, cm^{-2}$ UV 254 nm, and the other batch was kept on ice. The PNK assay was performed according to Holmqvist et al.[37]. Briefly, the cell pellets were dissolved in NP-T buffer (50 mM $Na_2HPO_4$ pH 8.0, 300 mM NaCl, 0.05% Tween 20), and bacteria were disrupted with 0.1 mm glass beads (Roth) on Fastprep-24 5G (MP Biomedicals). The lysates were cleared by centrifugation and the supernatants were collected. The cross-linked protein-RNA complexes were immuno-precipitated using the Anti-FLAG M2 Magnetic Beads (Merck). The RNA was partially degraded by incubation with 1:1000 dilution of 10 U/µL RNase I (Thermo Scientific) for 3 min at 37 °C. The RNA fragments were dephosphorylated with 10 U of CIAP (Thermo Scientific) for 30 min at 37 °C. The magnetic beads of both UV− and UV+ samples were separated into two batches: PNK− and PNK+. The PNK+ samples were labelled with γ−32P ATP (Hartmann Analytic) using T4 polynucleotide kinase (Thermo Scientific), while in the PNK− reaction, the enzyme was not included. The protein-RNA complexes were eluted from beads with 0.2 mg/mL 3x FLAG peptide (Merck). The eluted protein-RNA complexes were resolved on NuPAGE 4–12% Bis-Tris Protein Gel with MOPS buffer (Thermo Scientific) and transferred to the 0.45 µm nitrocellulose membrane (Cytiva). The radioactive signal was detected using the Typhoon FLA 9500 imager (Cytiva). Western blotting was then performed using 1:5000 Monoclonal ANTI-FLAG M2 antibody produced in mouse (Merck; F1804).

### Bioinformatic analysis of YebC proteins

The structure of *S. pyogenes* YebC was modelled on the basis of CBU_1566 from *Coxiella burnetii* (PDB 4F3Q) with SWISS-MODEL webserver[75]. The structure was visualised with PyMOL Molecular Graphics System, Version 3.0 (Schrödinger, LLC). The electrostatics calculations were performed using the APBS software[76]. The amino acid sequences of *S. pyogenes* YebC (SPy_0316) and its homologues were downloaded from the KEGG database[32]. The sequences were aligned with Clustal Omega[77]. The alignment and secondary structures of SPy_0316 and *Mus musculus* TACO1 (PDB 5EKZ) were visualised using ESPript[78]. The distribution of YebC_I and YebC_II subtypes in bacterial species was extracted from the PANTHER database[79] and visualised using the iTOL browser[80].

### RNA sequencing and MS proteomics of *S. pyogenes* Δ*yebC* strain

The WT, Δ*yebC* and Δ*yebC / yebC*+ strains (Supplementary Table 3) were grown in THY until the mid-logarithmic and early stationary phases. For RNA-seq, 10 mL of cell cultures were mixed with 5 mL of Stop solution (5% phenol in absolute ethanol) and cooled in an ice-water bath. For MS proteomics, 10 mL of cell cultures were cooled in an ice-water bath. The cells were precipitated by centrifugation for 5 min at $4500 \times g$ +4 °C. The pellets were frozen and stored at −80 °C. The cells were disrupted with 0.1 mm glass beads (Roth) on Fastprep-24 5G (MP Biomedicals).

RNA was isolated by acid guanidinium thiocyanate-phenol-chloroform extraction. To prepare RNA-seq libraries, the RNA was treated with Turbo DNase (Thermo Scientific) and purified using the RNA Clean & Concentrator-5 kit (Zymo Research). Ribosomal RNA was depleted with Pan-Bacteria riboPOOL kit (siTOOLs). 40 ng of RNA of each sample were used to create the libraries using the NEBNext Ultra II Directional RNA Library Prep Kit (New England Biolabs). Sequencing of the libraries was performed on NovaSeq 6000 (Illumina) in a $2 \times 100$ bp paired-end mode. Reads were filtered with a minimum quality score of 10 and a length of at least 18 nt. The reads were cleaned from adapter sequences using Cutadapt (v4.4)[81] and mapped against the reference genome NC_002737.2 using STAR (v2.7.3a)[82] in 'random best' and 'end-to-end' modes. BAM files were sorted and indexed using Samtools (v1.19.2)[83], and PCR duplication artefacts were removed using UMI Tools (v1.1.5)[84]. Gene counts of annotated RefSeq genes were determined with featureCounts (v2.0.1)[85] and differentially expressed genes (DEG) were calculated with three replicates per condition and growth phase using DESeq2 (v1.38.0)[86]. A threshold of adjusted $p < 0.05$ and absolute log2 fold change >1 was used to call DEGs. We performed the aforementioned analysis steps using a customised pipeline implemented in Snakemake (v8.4.8), a workflow management system[87].

For Data-independent acquisition (DIA) proteomics, the lysates were cleared by centrifugation. 10 µg of a 1:1 mixture of hydrophilic and hydrophobic carboxyl-coated paramagnetic beads (SeraMag, GE Healthcare) was added for each µg of protein. Protein binding was induced by the addition of acetonitrile to a final concentration of 70% (v/v). Samples were incubated for 10 min at room temperature. The tubes were placed on a magnetic rack, and beads were allowed to settle for 3 min. The supernatant was discarded, and beads were rinsed thrice with 80% ethanol. Beads were resuspended in a digestion buffer containing 50 mM triethylammonium bicarbonate (Sigma) and Trypsin and lysC (SERVA) in a 1:50 enzyme-to-protein ratio. Protein digestion was carried out for 14 h at 37 °C. The peptide supernatant was then recovered and acidified with 2% ACN and 0.1% trifluoroacetic acid.

Label-free DIA analyses of peptides were acquired over 120 min by an Orbitrap Exploris 480 (Thermo Scientific) coupled to a 3000 RSLC nano UPLC (Thermo Scientific) from 1 µg of peptides. Samples were loaded on a pepmap trap cartridge (300 µm i.d. × 5 mm, C18, Thermo Scientific) with 2% acetonitrile, 0.1% TFA at a flow rate of 20 µL/min. Peptides were separated over a 25 cm analytical column (PepSep C18, 75 µm I.D., 1.5 µm). Solvent A consists of 0.1% formic acid in water. Elution was carried out at a constant flow rate of 250 nL/min within 120 min. Initially, a two-step linear gradient was applied: 3–30% solvent B (0.1% formic acid in 80% acetonitrile) within 70.5 min, 30–45% solvent B within 13 min, followed by column washing and equilibration. The column was kept at a constant temperature of 50 °C. The MS was operated in DIA mode for single-injection quantitative measurements of individual samples with the following settings: 60k MS1 resolution, MS1 scan range 375–1400 m/z, 15k MS2 resolution, MS2 scan range 120–1600 m/z, Normalised AGC target of 1000%, maximum injection time 54 ms, and fixed normalised collision energy of 30. 12 m/z precursor isolation windows with optimised window placements from 400.4319 to 1204.7975 m/z.

Raw data analysis was performed using Spectronaut (Biognosys AG, Zurich, Switzerland) version 17 in directDIA+ deep mode with reviewed UniProt databases (*Streptococcus pyogenes* serotype M1 – version 2024_02_05, 1690 canonical entries). Methionine oxidation

and Acetyl (Protein N-term) were set as a variable, and carbamido-methylation on cysteine residues was used as a static modification. The FDR for PSM-, peptide-, and protein-level was set to 0.01. All tolerances were set to dynamic for pulsar searches. Extracted features were exported from Spectronaut for statistical analysis with MSstats[88] using default settings. Briefly, for each protein, features were log-transformed and fitted to a mixed-effect linear regression model for each sample. The model estimated the fold change and statistical significance for all compared conditions. The Benjamini−Hochberg method accounted for multiple tests, and a p-value adjustment was performed on all proteins that met the fold-change cut-off. Significantly different proteins were determined using a threshold of absolute log2 fold change >1 and an adjusted $p < 0.05$.

### Infection of human macrophages with *S. pyogenes*

For the generation of primary human macrophages, PBMCs were purified from buffy coats derived from the German Red Cross Blood Transfusion Service. Briefly, buffy coats were diluted 1:1 in PBS and separated in Pancoll density gradients (1.077 g/ml, Pan Biotech; 800 g, 30 min, room temperature, without break). PBMCs were washed twice with PBS and plated at a density of $1 \times 10^7$ cells in RPMI in 10 cm tissue culture dishes (Sarstedt) for 3 h. Then the medium was replaced by RPMI supplemented with 10% human serum type AB and 100 ng/mL GM-CSF (Miltenyi Biotec). The following day, the cells were washed once with PBS and further incubated in RPMI supplemented with 10% human serum type AB and 100 ng/mL GM-CSF for 5 days in a humidified atmosphere containing 5% $CO_2$. Media was replaced for RPMI + 10% human serum, type AB without GM-CSF and macrophages were incubated for a further day before being plated in 24-well plates at a density of $2 \times 10^5$ cells per well.

Infection assays were performed with *S. pyogenes* from mid-logarithmic phase cultures in THY ($OD_{600\ nm}$ - 0.5). Bacteria were centrifuged ($4700 \times g$, 5 min), washed with PBS and passed ten times through a 27 Gauge syringe insert in a 1 ml syringe (Braun) to separate the cocci chains. Cells were infected for 1 h with a Multiplicity of Infection (MOI) of 5 in RPMI + 1% human serum including centrifugation for 5 min at $300 \times g$ for synchronisation. For gentamicin protection assays, the macrophages were subsequently incubated with RPMI + 10% human serum and 100 µg/mL Gentamicin for 1 h to eliminate extracellular bacteria. The macrophages were further incubated with 10 µg/mL gentamicin until the end of the experiment. To determine intracellular bacterial loads, the macrophages were lysed at the indicated time points using 0.1% Saponin in PBS, and serial dilutions of the lysates were plated on THY agar plates and incubated for 24 h at 37 °C with 5% $CO_2$. For LDH and ELISA experiments, following the 1 h infection period, the macrophages were incubated with RPMI + 10% human serum and 1% penicillin/streptomycin. Supernatants were harvested at 4 h and 24 h post-infection.

Cytokines and LDH in macrophage supernatants were measured by R&D Systems DuoSet ELISA Development Systems (human IL-1β, DY201; human IL18, DY318; human IL-6, DY206; human TNF, DY210; human IL-8, DY208) or using the LDH Assay Kit (Abcam) according to the manufacturer's instructions. The percentage of cytotoxicity was calculated as follows:

$$cytotoxicity(\%) = \frac{100 \times (infected\ sample - low\ control)}{high\ control - low\ control}$$

### iCLIP for YebC in *S. pyogenes*

*S. pyogenes* wild type and *yebC::3xFLAG* strains were grown in CDM until the mid-logarithmic phase. The cells were rapidly cooled in an ice-water bath and collected by centrifugation. The cells were re-suspended in the ice-cold PBS and UV irradiated with 600 mJ cm$^{-2}$ UV 254 nm light. The cells were collected by centrifugation. Growth

and UV irradiation of *yebC::3xFLAG* cultures ('YebC_UV+') were performed in two biological replicates. The UV irradiated wild type cells represent the 'WT_UV+' control. An additional *yebC::3xFLAG* culture was not treated with UV and represents the 'YebC_UV−' control.

The iCLIP experiment was performed according to Buchbender et al.[42] with bacteria-specific modifications adapted from Andresen et al.[89]. Briefly, the cell pellets were re-suspended in NP-T buffer (50 mM $Na_2HPO_4$ pH 8.0, 300 mM NaCl, 0.05% Tween 20), and bacteria were disrupted with 0.1 mm glass beads (Roth) on Fastprep-24 5 G (MP Biomedicals). The lysates were cleared by centrifugation, and RNA was partially degraded by incubation with 1:25000 dilution of 10 U/µL RNase I (Thermo Scientific) for 3 min at 37 °C. At this step, portions of UV irradiated lysates of *yebC::3xFLAG* strain were collected and served as 'size-matched input' (SMI) controls. The cross-linked protein-RNA complexes were immunoprecipitated using the Anti-FLAG M2 Magnetic Beads (Merck). The RNA fragments were dephosphorylated and labelled with γ−32P ATP (Hartmann Analytic) using T4 polynucleotide kinase (Thermo Scientific). The protein-RNA complexes were eluted from beads with 0.2 mg/mL 3x FLAG peptide (Merck).

The YebC_UV+ samples and controls (WT_UV+, YebC_UV− and SMI) were resolved on NuPAGE 4-12% Bis-Tris Protein Gel with MOPS buffer (Thermo Scientific) and transferred to the 0.45 µm nitrocellulose membrane (Cytiva). The radioactive signal was detected using the Typhoon FLA 9500 imager (Cytiva). The zones corresponding to the labelled RNA fragments in the YebC_UV+ samples (~ 30−65 kDa) were excised from the membrane using the phosphor imaging print-out as a mask. The RNA was isolated from the pieces of nitrocellulose membrane and ethanol precipitated. The SMI RNA was dephosphorylated using T4 polynucleotide kinase and purified with the Oligo Clean & Concentrator kit (Zymo Research). L3-App oligo (Supplementary Table 1) was ligated to all RNA samples with T4 RNA Ligase 2, truncated K227Q (New England Biolabs) in the presence of 12% PEG8000 at 25 °C for 2 h. RNA was resolved on 12% Urea TBE PAGE and the RNA fragments with the lengths 40−300 nt were cut and eluted from the gel. RNA was ethanol precipitated with GlycoBlue Coprecipitant and reverse transcription was performed with SuperScript III (Thermo Scientific) and RToligo primer. Afterwards, RNA was degraded with 0.2 M NaOH by incubation at 65 °C for 15 min. cDNA was purified with Oligo Clean & Concentrator kit. The second adaptors L07-L12clip2.0 were ligated with T4 RNA Ligase 1, High Concentration (New England Biolabs) in the presence of 15% PEG8000 at 16 °C overnight. Afterwards, the cDNA libraries were purified using MyONE Silane beads (Thermo Scientific) and amplified by a two-round PCR with Phusion DNA polymerase (Thermo Scientific). The first 10-cycle PCR was performed with primers P5Solexa_s and P3Solexa_s. The PCR products were resolved on 12% native TBE PAGE, the fragments 75−300 bp were excised and eluted from the gel. After ethanol precipitation, they served as a template for the second round of PCR with primers P5Solexa and P3Solexa. For the SMI control, 9−11 cycles of the PCR were performed, and for the other samples and controls, 16 cycles. The amplified libraries were purified using the ProNex Size-Selective Purification System (Promega) at a sample-to-ProNex (vol/vol) ratio of 1:2.2. Sequencing of the libraries was performed on the NovaSeq 6000 (Illumina) in a 2 × 100 bp paired-end mode using the Seq_1 and Seq_2 primers.

iCLIP reads were pre-processed as described in Bush et al.[44]. In brief, reads with a high quality (minimum quality score of 20) in the barcode region were filtered using FASTX-Toolkit (v0.0.13) and seqtk subseq (v1.3), demultiplexed and trimmed using Flexbar (v3.5.0)[90]. Reads with a length of at least 18 nt were then mapped against the reference genome NC_002737.2 using STAR (v2.7.10a_alpha_220314) in 'random best' and 'Extend5pOfRead1' modes. The BAM files were sorted and indexed using Samtools (v1.9), and PCR duplication artefacts were removed using UMI Tools (v1.1.0). For each sample, transcriptional coverage was calculated with custom R scripts using the

GenomicRanges package[91]. Nucleotides located directly upstream of the mapped reads were considered cross-linking sites and for each genomic position, the number of the cross-linking sites was calculated. The coverage and the number of the cross-linking sites at each genomic position were divided by the total number of mapped reads and multiplied by $1 \times 10^6$ to obtain the normalised transcripts per million (TPM) values. The normalised coverage and the density of cross-linking sites of six *S. pyogenes* rRNA operons were collapsed to the first operon (SPY_RS00070, SPY_RS00080 and SPY_RS00085). Using a sliding window approach, the peaks of the cross-linking sites were mapped: these are 30 nt long regions where the sum of TPMs for the cross-linking sites is greater than three and more than five times greater than that of the 30 nt long up- and downstream regions. For visualisation of the cross-linking sites in 23S rRNA, a molecular model of the *E. coli* ribosome (PDB ID 7K00) was used to create a molmap at 3 Å in UCSF ChimeraX 1.8[92].

### Association of YebC with ribosomes

The *S. pyogenes* strain *yebC::3xFLAG* was grown in THY until the mid-logarithmic and stationary phases. The cells were collected by rapid filtration, immediately frozen in liquid nitrogen and stored at −80 °C. The cells were disrupted on MM400 swing mill (Retsch) in the presence of 1 mM chloramphenicol. The cell powders were collected and melted on ice. The lysates were cleared by centrifugation and applied to the 10–50% sucrose gradients in SW40 ultracentrifuge tubes (Beckman Coulter). The tubes were centrifuged in SW 40 Ti Swinging-Bucket Rotor (Beckman Coulter) at 35,000 rpm for 2 h at 4 °C. The gradients were fractionated with Piston Gradient Fractionator (Biocomp) equipped with FC 203B Fraction Collector (Gilson). During the fractionation, $OD_{260 nm}$ measurements were performed using the Triax Flow Cell (Biocomp). The proteins from the fractions were precipitated with trichloroacetic acid and resolved on Laemmli SDS-PAGE gels. One of the gels was stained with Coomassie. Proteins from the other gel were semi-dry transferred to a 0.45 μm NC membrane and western blotting was performed using 1:5000 dilution of Monoclonal ANTI-FLAG M2 antibody produced in mouse (Merck; F1804). Alternatively, the in vitro translation reaction with P5 mRNA as a template and the YebC_II (b1983) protein was carried out for 40 min. The reaction was cooled on ice and resolved on the sucrose gradient. The fractions were collected and analysed by western blotting with a 1:5000 dilution of the HA Tag Monoclonal Antibody (Thermo Scientific; #26183).

### Ribosome profiling for *S. pyogenes yebC* mutants

The ribosome profiling protocol is based on Galmozzi et al.[93] and was adapted to *S. pyogenes*. The WT, Δ*yebC*, Δ*yebC/yebC*+ and Δ*yebC/yebC_M2* strains were grown in THY until the mid-logarithmic phase. The cells were collected and disrupted in the same way as for the study of the association of YebC with ribosomes. mRNA was partially degraded with Nuclease S7 (Merck). The lysates were resolved on 10–50% sucrose density gradients and the monosome fractions were collected. RNA was isolated by phenol-chloroform extraction and resolved by electrophoresis on the Urea TBE PAGE. The 15-45 nt long ribosome protected mRNA fragments were isolated from gel and ethanol-precipitated.

The RNA was dephosphorylated with T4 Polynucleotide Kinase (Thermo Scientific) in buffer A for 2 h at 37 °C. The RNA was precipitated with 2-propanol and GlycoBlue Coprecipitant. The pre-adenylated adaptor with unique molecular identifier (Supplementary Table 1) was ligated with T4 RNA Ligase 2, truncated K227Q (New England Biolabs) in the presence of 25% PEG8000 at 25 °C for 2 h. The adaptor was also ligated to the mix of 15 and 45 nt control RNAs and this sample later served as guides for gel-purification. In order to remove the non-ligated adaptor, Rec J exonuclease (Biosearch Technologies) and yeast 5′-deadenylase (New England Biolabs) were added and the reactions were incubated at 30 °C for 45 min. Reverse

transcription was performed with SuperScript III (Thermo Scientific) and RT primer and the RNA was degraded with 0.2 M NaOH by incubation at 65 °C for 15 min. cDNA fragments were purified using the Oligo Clean & Concentrator kit and resolved on 10% Urea TBE PAGE. Using the size guides, the fragments with 15−45 nt inserts were excised and isolated from the gel. cDNAs were precipitated with 2-propanol and GlycoBlue Coprecipitant and re-dissolved in 15 μL of mQ. Circularisation of the cDNA molecules was performed with CircLigase II ssDNA Ligase (Biosearch Technologies) for 1 h at 60 °C. The cDNA libraries were then amplified with primers introducing the sequencing barcodes. The amplified libraries were purified with the ProNex Size-Selective Purification System (Promega) at a sample-to-ProNex (vol/vol) ratio of 1:2.2. The libraries were sequenced on NovaSeq 6000 (Illumina) in a 100 bp single-end mode.

To obtain the ribosome profiling footprints, reads were filtered for a minimum quality score of 10 and a length between 22 and 52 nt, cleaned from adaptor sequences using Cutadapt (v4.4) and mapped against the reference genome NC_002737.2 using STAR (v2.7.3a) in 'random best' and 'end-to-end' modes. BAM files were sorted and indexed using Samtools (v1.6). Random unique molecular identifiers that were introduced at the 5′ end (2 nt) and 3′ end (5 nt) of the footprint were detected and PCR de-duplication was performed using UMI Tools (v1.1.4). Reads mapping to rRNA or tRNA features as defined in the NCBI RefSeq annotation of NC_002737.2 were removed using Bedtools (v2.31.0)[94]. To obtain these pre-processed and mapped ribosome footprints, we executed our customised ribo-seq snakemake pipeline until the 'filter_bam' step in the preprocessing module.

For all subsequent analyses, the reads having length between 27 and 40 nt were selected. The position of the ribosomal P site was mapped 15 nt upstream of the 3′ ends of the reads[47,48] and for each codon of *S. pyogenes* proteins, the number of reads representing the ribosomal A site was calculated with custom R scripts using the GenomicRanges package. To account for the difference in mRNA abundance, this number was divided by the sum of the numbers for the coding sequence. The resulting values were termed 'pause scores' and reflect the occupancy of each codon by the translating ribosome. To identify codons for which pausing depends on the presence of YebC, six replicates of YebC+ samples (*WT* and Δ*yebC/yebC*+ strains) were compared with six replicates of YebC− samples (Δ*yebC* and Δ*yebC/yebC_M2*). To estimate the statistical significance of the difference, we used the edgeR software[95], treating each codon as a gene and each coding sequence as a library. Pausing at the codons having an adjusted $p < 0.001$, a difference of pause scores greater than 4 and an average pause score in the up-regulated sample greater than 2 was considered to be affected by the mutation of YebC protein.

### Ribosome stalling reporter in *S. pyogenes*

The reporter strains (Supplementary Table 3) were grown in THY until the mid-logarithmic phase. Expression of the reporter protein was induced by addition of 10 ng/mL anhydrotetracycline and the cells were grown for 30 min. The cell cultures were cooled in an ice-water bath and the cells were collected by centrifugation for 5 min at $4500 \times g$ +4 °C. The bacteria were disrupted with 0.1 mm glass beads (Roth) on Fastprep-24 5G (MP Biomedicals). The lysates were cleared by centrifugation for 10 min at +4 °C 10,000 g and the supernatants were collected. The lysates were resolved on a NuPAGE 4-12% Bis-Tris Protein Gel with MOPS buffer (Thermo Scientific) and transferred to the 0.45 μm nitrocellulose membrane (Cytiva) using the semi-wet transfer apparatus (Thermo Scientific). Western blotting was performed using 1:5000 dilution of Monoclonal ANTI-FLAG M2 antibody produced in mouse (Merck; F1804). Expression of the full-length fusion proteins was measured by densitometry of western blot photograph with Fiji[96], and for each strain was normalised to the expression of P0 variant.

## Purification of *E. coli* YebC proteins

The coding sequences of *E. coli* *yebC* paralogs *b1983* (*yebC_II*) and *b1864* (*yebC_I*) were amplified from genomic DNA of strain BW25113 with primers listed in Supplementary Table 1. The expression vector pET-21a(+) was amplified with primers excluding the T7 tag sequence. HiFi DNA assembly (New England Biolabs) was used to assemble the expression vectors. The HA tag was then introduced to the C-termini of the genes by the whole-plasmid PCRs and circularisation with the KLD mix (New England Biolabs). The mutations M2 and Y84A were introduced to the *b1983* sequence by the whole-plasmid PCRs and circularisation with KLD mix. The introduction of M2 required two cycles of mutagenesis.

The plasmids pET-21a_b1983-HA, pET-21a_b1983-mut2-HA, pET-21a_b1983-Y84A-HA and pET-21a_b1864-HA (Supplementary Table 2) were introduced by transformation into *E. coli* BL21(DE3) cells (Thermo Scientific). The colonies were inoculated into 200 mL of LB medium with 100 μg/mL of carbenicillin and the cells were grown at 37 °C with 180 rpm shaking until $OD_{600\,nm} = 0.6$. The cultures were cooled on ice, the protein expression was induced with 200 μM IPTG overnight at 25 °C with shaking at 180 rpm. The cells were collected by centrifugation and stored at −80 °C. The cells were lysed with the B-PER Complete Bacterial Protein Extraction Reagent (Thermo Scientific) supplemented with 500 mM NaCl, 20 mM imidazole and cOmplete, EDTA-free Protease Inhibitor Cocktail (Merck). The proteins were immobilised on HisTrap HP columns (GE Healthcare), washed with buffer A (50 mM Tris-HCl pH 7.4, 500 mM NaCl, 20 mM Imidazole) and eluted with buffer B (50 mM Tris-HCl pH 7.4, 500 mM NaCl, 500 mM Imidazole) using a linear gradient on the ÄKTA pure chromatography system (Cytiva). The proteins were further purified by gel-filtration on Superdex 200 13/300 increase columns (Cytiva) in SEC buffer (25 mM Tris-HCl pH 7.4, 200 mM NaCl, 0.5 mM EDTA, 2 mM DTT, 10% glycerol). The purified proteins were analysed by Coomassie-stained SDS-PAGE, aliquoted and stored at −80 °C.

## In vitro ribosome stalling reporter

The templates for in vitro translation were designed on the basis of the DHFR control plasmid from the PUREexpress kit (New England Biolabs). The FLAG tag sequence and mutations were introduced to the *folA* N-terminus by two rounds of the whole-plasmid PCR with primers listed in Supplementary Table 1, and circularisation with KLD mix. The DHFR plasmids with the mutant versions of *folA* were linearised with the NotI FD restriction enzyme (Thermo Scientific) and used as templates for in vitro transcription using the MEGAscript T7 Transcription Kit (Thermo Scientific). The synthesised mRNAs were purified with the RNeasy Mini kit (Qiagen). In vitro translation was performed with the PUREExpress in vitro protein synthesis kit (New England Biolabs). The 10 μL PUREExpress reactions with or without 2 μM YebC variants were preincubated at 37 °C for 5 min. mRNA templates were heated at 70 °C for 1 min, cooled on ice and added to the reactions to a concentration of 1 μM. The reactions proceeded for 1 h at 37 °C. Next, 0.2 mg/mL RNase A (Qiagen) was added to some reactions and the RNA was degraded for 10 min at 37 °C. 2 μL of the reactions were resolved on a NuPAGE 4–12% Bis-Tris Protein Gel with MES buffer (Thermo Scientific) and transferred to the 0.2 μm nitrocellulose membrane (Cytiva) using the semi-wet transfer apparatus (Thermo Scientific). Western blotting was performed with 1:5000 of Monoclonal ANTI-FLAG M2 antibody produced in mouse (Merck; F1804).

## Quantitative RT-PCR

*Salmonella* Typhimurium strains were grown in LB medium until $OD_{600} = 0.6$ and aliquots of the cultures were mixed with RNAprotect Bacteria Reagent (Qiagen). RNA was isolated using RNeasy Mini kit (Qiagen) and treated with TURBO DNA-free Kit (Thermo Scientific). qRT-PCR was performed with primers listed in Supplementary Table 1 using the Power SYBR Green RNAto-CT 1-Step Kit (Thermo Scientific)

in the QuantStudio 5 Real-Time PCR System (Thermo Scientific) using 10 ng of DNase-treated RNA per 20 μL reaction. *gmk* and 16S rRNA served as reference genes, and relative expression was calculated as previously described[97,98].

## Reporting summary

Further information on research design is available in the Nature Portfolio Reporting Summary linked to this article.

## Data availability

All next generation sequencing data have been deposited in the European Nucleotide Archive (ENA) under the accession PRJEB78417. Processed alignment files of the ribosome profiling footprints and the iCLIP data have been deposited in the Open Research Repository (EDMOND) of the Max Planck Society [https://doi.org/10.17617/3.8PZNYF]. The proteomics mass spectrometry data have been deposited to the ProteomeXchange Consortium via the PRIDE[99] partner repository with the dataset identifier PXD054642. Source data are provided with this paper.

## Code availability

The source code to reproduce the main results reported in this study are available at https://github.com/MPUSP/ignatov_et_al_2025 [https://doi.org/10.5281/zenodo.15401960]. The source code for the ribo-seq pipeline is available at https://github.com/MPUSP/snakemake-bacterial-riboseq [https://doi.org/10.5281/zenodo.15403357].

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

## Acknowledgements

We thank Thomas Wulff (MPUSP) for providing sfGFP-mKate fusion, and Florian Kondrot and Matthias Münzner (MPUSP) for technical support. We are grateful to Felix Gersteuer (Laboratory of Daniel Wilson, Institute of Biochemistry and Molecular Biology, University of Hamburg, Germany) for preparing an illustration. We are also grateful to Kürşad Turgay (MPUSP) and Rainer Nikolay (Max Planck Institute for Molecular Genetics, Berlin) for scientific discussions and critical reading of the manuscript. This work was funded by the Max Planck Society (to E.C.), the German Research Foundation (DFG; Leibniz Prize to E.C.) and in part by a project from the European Research Council under the European Union's Horizon 2020 research and innovation programme (grant agreement n∘864971) and a Max Planck Fellowship of the Max Planck Society (to M.E.).

## Author contributions

D.I.: Conceptualisation, Methodology, Investigation, Formal Analysis, Software, Visualisation, Writing—review & editing; V.S.: Resources, Methodology, Investigation; R.A.: Data Curation, Formal Analysis, Software, Visualisation, Writing—review & editing; K.A.: Methodology, Investigation, Formal Analysis; K.H.: Investigation, Formal Analysis; C.W.: Resources, Investigation. K.K.: Resources, Investigation, Formal Analysis; F.A.C.: Investigation, Formal Analysis; K.F.: Investigation, Formal Analysis; M.E.: Resources, Methodology, Investigation; C.K.F.: Methodology, Investigation, Formal Analysis; E.C.: Conceptualisation, Funding Acquisition, Project Administration, Writing—Review & Editing.

## Funding

## Competing interests

The authors declare no competing interests.
