## [Transparent Peer Review file · Nature Communications]

RNA-binding protein YebC enhances translation of proline-rich amino acid stretches in bacteria

Corresponding Author: Professor Emmanuelle Charpentier

Version 1:

Reviewer comments:

Reviewer #1

(Remarks to the Author)

The manuscript by D. Ignatov and colleagues reports an approach developed for the identification of RNA binding proteins (RBPs) and experiments that support for the characterization of the newly identified RBPs. The authors provide compelling evidence that their approach successfully identified a number of proteins that interact with RNA in *S. pyogenes*. In the second half of the manuscript, the authors focused on one of the identified proteins, YebC. Deletion of *yebC* reduced the expression of the major *S. pyogenes* virulence factor, SpeB, and survival in human macrophages. Global mapping of the binding sites for YebC-3xFLAG suggested that YebC is a translational factor, and using the ribosome profiling as well as in vivo and in vitro translational reporter assays, YebC were demonstrated to promote the efficient translation of proteins containing polyproline stretches. This study nicely combines global analyses and biochemical assays, providing a convincing case for YebC's function for the efficient translation of proline-rich regions that will be of interest to researchers studying translation mechanism. The authors describe the methods in sufficient detail. A couple of points that are listed below seem important to further clarify, which the manuscript could be strengthened.

1. In Fig. 4C, the authors only show the secondary structure of part of 23S rRNA in order to map the location of the crosslinking region with YebC. There are structures of the ribosome available, and the peptidyl transferase center seem conserved among bacterial ribosomes. Hence, one of the ribosome structures should be used for the discussion to get better idea of the action of YebC in the translation of proline-rich regions.
2. In Fig. 6C, the decrease in the whole-proteome level of the *yebC* mutant was slight (statically significant in the log phase). In this proteome analysis, did the authors use the information of full-length proteins? Given the YebC affects translation elongation at certain positions, the comparison of N-terminal and C-terminal portions in each protein would help in accurately understanding the contribution of YebC.
3. In Supplementary Fig 7, interaction of YebC with the ribosome was not observed. In Fig. 6D, an in vitro translation system clearly demonstrated the YebC contribution to the translation of proline-rich regions. This pure condition would be appropriate to investigate the interaction of YebC with the ribosome. Can the authors show the polysome profile (or pull-down by YebC-3xFLAG) of the in vitro translation complex on the P5 reporter mRNA?
4. In discussion (lines 400-412, 432-442), the authors point out differences in mechanisms for YebC and EF-P, but they also refer previous studies that showed phenotypes caused by YebC, some of which are likely relevant to the EF-P function. Does the *S. pyogenes* genome have the *efp* gene? If this is the case, did the authors investigate phenotypes, such as growth curve (Supplementary Fig 5A), of an *efp* mutant or a *yebC efp* double mutant?
5. Related to Q4, in Fig. 2D the authors showed the phylogenetic distribution of YebC family proteins among bacterial genomes. To compare this with that of EF-P (or YfmR) may help in understanding the roles of these translational factors in the translation.
6. In Supplementary Fig 5A, the *yebC* deletion increased the cell growth at the stationary phase under chemically defined medium condition, but not under the other medium conditions. Did the authors compare the expression level of *yebC* under these culture conditions?

7. In lines 446-448, the authors referred that the expression of SpeB is affected by a quorum sensing and the environmental pH. In the transcriptome and proteome analyses of the yebC mutant, were the factors that sense these environmental cues separately investigated? Does "its regulator" in line 447 indicate the factor involved in quorum sensing or pH modulation?

Reviewer #2

(Remarks to the Author)

In the manuscript entitled "Novel RNA-binding protein YebC enhances translation of proline-rich amino acid stretches in bacteria" submitted to Nature Communications, Ignatov and colleagues sought to identify and investigate novel RNA binding proteins in group A Streptococcus, an important human pathogen. Using the global approaches OOPS and RBS-ID, the authors discovered a set of previously unreported, putative RNA binding proteins, in addition to established ones (Fig. 1 and Table 1) in *S. pyogenes*. The authors initially focused their efforts on five putative RNA binding proteins assessing their ability to interact with RNA in vivo and found via the immunoprecipitation of crosslinked RNA complex approach that YebC, PhoH, ThuC, YjbK and YgaC interact with RNA (Fig. 2). The authors then concentrated on YebC, which is highly conserved in bacteria and eukaryotes, but little was known regarding its function until very recently. Through deletion and mutation analyses of yebC, the authors provide evidence that YebC is important for *S. pyogenes* survival in macrophages and expression of the virulence factor SpeB. The authors also established the role of particular domains and residues in YebC function (Fig. 3). Using an exhaustive set of high throughput, molecular, and biochemical approaches, the authors determined that YebC interacts with the 23S rRNA and assists the ribosome in translating through stretches of regions of mRNA encoding multiple proline residues (Figs. 4, 5, and 6).

Overall, this manuscript is well written and details a thorough investigation of RNA binding proteins in *S. pyogenes* and in particular YebC, establishing a detailed molecular mechanism of the function of this protein that is highly conserved in at least two domains of life. The overwhelming amount of data presented by the authors resolutely support their model of YebC function. While there is a manuscript recently published in NAR that identified the human homologue TACO1 in assisting the ribosome in translating through proline rich codons, it is my opinion that this work only slightly detracts from this manuscript under review. This manuscript not only validates the conclusions of that work by Brischigliaro et al (PMID 39036954), but also establishes additional details of the molecular mechanism of YebC demonstrating that it interacts with a specific region of the 23S rRNA, establishing the role of distinct domains and residues in its function, and demonstrating the impact of this protein on *S. pyogenes* in terms of its growth, virulence, and proteome. Additionally, the OOPS and RBS ID data generated from this work revealed many other putative RNA binding proteins, and this data is useful for others investigating RNA binding proteins and RNA metabolism in both closely and distantly related bacteria that have homologous proteins. In summary, this is an exciting, highly impactful manuscript.

Major Criticisms:

NONE

Minor comments:

Fig. 4A legend. Although it is defined in the methods section, it would have been useful to me and perhaps others, if the authors defined SMI here as size matched input.

Fig. 4D. Since I and perhaps other readers are not familiar at looking at traces of ribosomal complexes, it might be helpful to provide some labeled axis for these results such as labeling an X-axis as sucrose concentration and Y-axis as signal intensity, unless some other convention exists.

References. The names of species are not italicized as is typical convention.

Reviewer #3

(Remarks to the Author)

The manuscript by Ignatov et al. describes the identification and characterization of YebC_{II} as a translation factor that facilitates translation of polyprolines in *Streptococcus pyogenes*. The authors identified YebC_{II} as an RNA binding protein following UV-cross linking and mass spectrometry. Indeed, this screen uncovered a number of novel RNA-binding proteins as well as re-confirming other known-RNA-binding proteins. Authors were also able to identify residues that are likely RNA-interacting on YebC as well as the region of the ribosome (Helix89 near the PTC) with which YebC is likely to interact. These data are very helpful in lieu of a structure. Authors follow up on the YebC residues of interest with site-directed mutagenesis. Finally, authors performed ribosome profiling to identify mRNA sequences with significant pause scores and found stalling at PP, PXP, or DXP motifs. Authors followed up using polyproline-encoding in vitro and in vivo reporters.

The study is thorough, well-executed, and very timely. Major claims are well supported by the data. Authors present a comprehensive mix of proteomics, biochemistry, genetics, ribosome profiling, and specific reporters to support their findings.

Specific comments for improvement:

1. Line 57: Authors should re-word the statement "YfmR/Uup has also been shown to promote translation of proline-rich

motifs in cooperation with EF-P”

This wording makes it sound like EF-P and YfmR work together to relieve proline stalling, but this is unlikely to be the case since binding of one would hinder binding of the other. Deletion of both factors indeed causes a severe growth defect with many orders of magnitude decreased survival, but all current evidence suggests EF-P and YfmR function independently on the ribosome.

2. Line 125: Authors should modify language surrounding the conservation of YebC. It is not clear that YebC is found in most bacteria. To say that the protein is broadly conserved would be more accurate, especially since the degree of conservation is different for each paralog. It is also not accurate to say it is universally conserved without the identification of an Archaeal homolog.

3. The identification and validation of residues involved in RNA-binding is very exciting. Can the presence or absence of these residues in the YebC_I family be used to speculate on their function? Are they conserved in YebC_I or absent? Either finding would be interesting since one exciting mystery is whether the YebC_I family proteins may also function in translation. A modest role for YebC_I in translation is also supported by Figure 6D, though it is not as significant as for YebC_II. Marking the important YebC_I residues in Figure S4 would be helpful.

4. Authors should reduce claim that EF-P and YfmR are absent in the PURE system. The PURE system certainly has a lot of additional ribosome-associating factors that co-purify with ribosomes. I do not find this problematic for the conclusions of the study, just an important thing to recognize about PURExpress, and particularly important because of the potential for YebC to bind the A-site. If that is the case, EF-P and YfmR/Uup could still bind the ribosome at the same time even if they aren't necessarily working together.

5. I am curious about increased optical density in stationary phase. What could be the reason for this? Have authors looked at mutant cells under the microscope? Does the cell volume increase in stationary phase?

6. The differential expression of *speB* remains mysterious. Although a significant pause-site was not identified in the ribo-seq data, are there are other features of *speB* that could be of interest for ribosome stalling? For example, prolines near the 5' or 3' end where the stalling may be less obvious due to increased ribosome occupancy from initiation or termination? Are there transcriptional regulators upstream of *speB* that exhibit ribosome stalling? It would be interesting to compare an in vivo and an in vitro *speB* reporter but not essential for the conclusions of the paper.

Minor:

Typo at line 21 – missing “an” before “in vitro system”.

Reviewer #4

(Remarks to the Author)

Review of Ignatov et al., “Novel RNA-binding protein YebC enhances translation of proline-rich amino acid stretches in bacteria”

The manuscript from Ignatov et al, entitled “Novel RNA-binding protein YebC enhances translation of proline-rich amino acid stretches in bacteria” describes a comprehensive proteome-wide study in *S. pyogenes*, to retrieve its RNA binding proteome (RBPome). The authors took two approaches to enrich the RBPome: OOPS, an approach using phase separation of UV crosslinked RNA-bound proteins, and RBS-ID, a method that uses hydrofluoride to fully cleave RNA into mono-nucleosides and allows RNA binding site mapping at single amino acids resolution. Using these approaches the authors were able to return a credible list of RBPs from *S.pyogenes*. They went on to delve into the function of one of these proteins, YebC, their data suggesting a role in facilitating translation at sites of polyproline in translation proteins.

This work is timely, insomuch that many large scale RBPome studies are available, but few go onto the interrogate the function of ‘novel’ RBPs present in their data. The study is mostly well executed, but lacks a certain novelty, as other studies have already assigned function to YebC in other organisms, and thus the main results of the study are not surprising. However, the data presented do represent the only RBPome data from this pathogenically important bacterium.

The manuscript does require some clarification for it to be suitable for publication. Moreover, a fuller description of existing publicly available datasets is also required to provide an up-to-date representation of the field.

Major Comments:

1. Novelty/contribution to the field: The authors have done a commendable job in identifying RBPs in *S. pyogenes*, but YebC has already been characterised in *E. coli* (Monti et al., 2024) and in human studies (as TACO1). This weakens the novelty of the central finding. The introduction would benefit from acknowledging these studies earlier, providing a more cohesive backdrop to YebC's conserved functions across species and why the authors focused on this protein.

2. Dataset Comparisons: The study's claim of novelty in identifying YebC and other RBPs would be stronger if the authors systematically compared their findings with other RBPome studies. Expanding the supplementary data to include orthology-based comparisons across species (including human and *E. coli* RBPomes) would significantly enhance the robustness of the findings.

3. Methodological points:

○ The UV crosslinking dose (600 mJ/cm²) used in the OOPS method is not justified in the manuscript. The authors should provide evidence of a dose-response experiment to support this choice, particularly as they are using a non-model organism or at least disclose why they chose this dose.

○ Clarification on the use/limitations of SILAC in *S. pyogenes* is required, as this bacterium synthesises its own amino acids, which may limit efficient isotope incorporation.

○ Figure 2B would strongly benefit/would be more convincing if they did a densitometry quantification between the UV+PNK+ vs UV-PNK+ at the minimum. The negative control (enrichment 0.01) looks like it has a stronger signal/shift compared to YgaC (enrichment 3.21) ... this might just be because it needs to be normalised by the protein amount in the WB. A barplot will make the values much more intuitive/the results more convincing/conclusive.

4. OOPS and RBS-ID Analysis:

○ The manuscript lacks a detailed interpretation of the physicochemical properties of the amino acids enriched in the RBS-ID experiment. Do the enriched sites match known RNA crosslinking motifs? Including this analysis would provide more confidence in the approach.

○ GO term analysis is insufficiently detailed, especially for bacterial systems. Integrating orthology information to identify conserved RBPs across species would be more meaningful than relying solely on GO term enrichment, which is limited for bacterial systems.

○ 'The annotated RBPs, the proteins with significant OOPS enrichment, and the 87 proteins with detected RNA-binding sites overlapped significantly (Fig. 1B)'. Figure 1B requires clarification of the significance of the overlap between different datasets (OOPS, annotated RBPs, and RBS-ID proteins). A statistical test (e.g., a contingency table) to quantify this overlap would be useful.

5. Clarification of Terminology:

○ The authors' use of the term "novel RBPs" is questionable. Many proteins described as novel RBPs in this manuscript have been identified in other bacterial species, even if uncharacterized. It would be more accurate to describe them as "uncharacterized" rather than "novel." (or some other term that the authors find more representative)

○ The interaction sites of YebC should be contextualised with respect to known RNA-binding domains (e.g., helix-turn-helix domains) to better link RBS-ID data with biological function.

Minor Comments:

● The introduction does not properly contextualise the field of RBPome in the bacterial world and/or explain why it has been delayed compared to the eukaryotes due to experimental limitations. "These studies have identified ProQ" is a very oversimplified summary of the yield of RBPome studies in this area. It would be to the benefit of the papers for the authors to highlight the relatively few manuscripts (in bacteria) which have functionally dissected the role of newly-identified RBPs. This paper is part of a larger community effort to start characterising the RBPome beyond the accumulation of catalogues, but the value of this needs to be made more clear in the introduction. Moreover, Monti et al also discusses YebC as a novel *E. coli* RBP and describe a mitochondrially localised human orthologue – they should acknowledge this in the introduction rather than leave it to the discussion. Finally, in the discussion they mention a paper in humans demonstrating the role of TACO1 (YebC ortholog) in resolving polyP in mitochondria, but this was not mentioned in the introduction.

● The supplementary tables are excellently labelled and very easy to understand/navigate, we thank the authors.

● We would recommend expanding Supplementary data 2 to include orthology-inferred RBPs- it's high time we start integrating all the RBP catalogues to get a more refined picture of what is going on instead of collections of stand-alone catalogues of 'novel' RBPs- I wouldn't be surprised if many of these 'novel' candidates actually appear to be consistently identified as RBP in other species. This would be significant and make the data more robust!

● On this note Supplementary figure 2 would be extremely informative to the field if it incorporated the RBP info from human, mouse, ... *E. coli* to see which candidates are constantly appearing.

● Further, the red and blue dots in Sup. Fig. 2 are not clear and are not labelled in the caption- they must be the OOPS enrichment and the detection of RNA-binding, but which is which?

● Additionally- would be intrigued to see which GO terms are found in the different categories of Fig 1B, e.g. are the proteins pulled out in RBS-ID only different functionally/physicochemically compared to those only seen in OOPS?

● 'Glycolysis enzymes from different species often function as non-canonical RBPs' I would suggest expanding the citations to primary research/RBPome studies that demonstrate and validate this clause instead of a review.

● Why did the authors not follow up YjbK, YgaC and ThuC? In the discussion there is some description of what the protein might do, but the reason not to follow these up further could have been better enunciated.

● 9 of these are detected as DNA-binding proteins- need to evaluate in the light of recent findings from Chu et al 2022 (<https://doi.org/10.1038/s41467-022-30553-8>) where they demonstrate that DNA-binding and Rossmann-fold proteins show RNA-binding. Are these domains detected in these proteins? I think a quick look to see if this domain hypothesis is true would be meaningful.

● Plot 2C is excellent- it makes me wonder- are the RBS-ID sites enriched in RNA-binding domains? It would be nice to see this analysis in the supplementary (!) and give more confidence to the approach

● 'Suggesting that it plays an important role in bacterial physiology' - no reference to the fact that in Monti et al 2024 the yebC KO the growth rate and saturation OD count was affected in both M9LB and under nutrient stress, perhaps a hypothesis to the matter can be explored in the discussion.

● The human ortholog (TACO1) has been detected as RNA-binding in human assays (Queiroz et al, 2019; Castello et al, 2016 - not sure about mice!) but has not been referred to here.

● Figure 2D: what is blue and red? Simply the species? So many of the species listed have an RBP dataset!! Please look, see if RNA-binding conserved, would make the conclusions so much more powerful!

● Figure 2E: Also note that YebC has a HTH domain (at least in *E. coli*) which as I mentioned above has been proven to bind RNA in vivo (Chu et al, 2022)- are the interaction sites in this domain? This would showcase the value of having both OOPS data AND RBS-ID to integrate and extract more meaningful/complete data.

● Why a C-terminal tag? Is this further (structurally) from the predicted binding region of the protein?

● Describe what SpeB is when it is first mentioned

● Confusing that Figure 3A is the same as Sup. Fig. 5C

● Do the mutations affect the RNA-binding ability? Could the authors take the yebC-/yebC+ variants for each mutant, apply OOPS and evaluate the WB signal of YebC in the organic vs the interphase? This would be essential to decouple: is the

capacity to bind RNA necessary for YebC function? Or instead, is RNA binding an additional function of YebC?

- In the results, a clearer explanation of the iCLIP data analysis, especially for non-expert readers, is necessary. Terms like "size-matched input libraries" should be defined, and a better explanation of the rationale behind 'we extracted the cross-linked nucleotides located immediately upstream of the mapped cDNA reads'.
- I am not sure they included the right control for iCLIP- you are irradiating the cell so RNA that is proximal to YebC will be covalently bound. Given the sheer abundance of ribosomes, there is a chance you are capturing noise... I think the most elegant would have been to do the experiment I suggest above of seeing which mutants are unable to bind RNA and use that 'RNA-binding-dead' mutant as a control. It could be that YebC interacts with the ribosome not necessarily the 23S rRNA. In any case this could be addressed with an EMSA of the YebC and the RNA-binding-dead YebC variant to avoid the cost of repeating iCLIP. This of course would only be the case if the authors obtain reproducible/convincing EMSA shifts already with YebC in WT state.
- 'Our search for RBPs in *S. pyogenes* identified most of the proteins involved in translation, including ribosomal proteins, proteins involved in ribosome biogenesis and translation factors. Therefore, it seems possible that YebC is also involved in ribosome biogenesis or translation. In order to investigate' - I do not see the immediate connection between these sentences here, need to be more explicit. Most RBPs will be translation related because those are the best annotated and most conserved (!) and possibly the easiest to capture due to their abundance and near constant binding to RNA.
- The distinction between the use of different media (THY for RNA-seq and CDM for KO characterization) needs to be clarified to avoid confusion.
- More emphasis on comparing the effects of YebC deletion with other known factors such as EF-P and YfmR would enhance the discussion of potential mechanisms underlying ribosome stalling.
- Supplementary fig. 8A, why only show the length distribution of one library? Can the authors comment on the distribution shown?
- 'Median pause scores were slightly higher when serine or glycine codons were located in the E site (Supplementary fig. 8C)' which statistics have been used, if any? Please clarify.
- Unclear what the x-axis of 5B is!
- 'To this end, we performed a proteome analysis of the same bacterial cultures that we used for the transcriptome analysis: WT, $\Delta yebC$ and $\Delta yebC$ / $yebC^+$ strains in mid-logarithmic and stationary growth phases (Supplementary data 7)' - missing to state the conclusion, where there any changes overall in the proteome in these pairwise comparisons? Visualising this in the supplementary. Would perhaps expect some changes if the phenotype is ribosome activity?
- For later in the discussion but why is 'ns' in the stationary phase in Fig 5C?
- An additional thought- is there any functional relevance to genes with polyP in this system? Would be interesting to do GO/KEGG on the candidate's pauses (and genes with polyP generally). If no terms appear significant, I would explore applying this on the Ecoli orthologs, as they may be better annotated. Perhaps a sliding approach is needed P1 -> P3 -> P5 given the changing significance depending on the extension of the polyP track shown in Fig 6
- 'We next used the PURE in vitro translation system' - after reading the discussion I understand this was done to exclude indirect effects of EF-P or YfmR but this is not immediately obvious to the non-expert reader. I would make the rationale more transparent already in the results section.
- In the discussion: 'This significantly narrowed the set of candidates for novel RBPs and filtered out the proteins that could stably interact with polysaccharides' - I think the rationale for this concern (contaminating interphase) is not immediately obvious to the non-specialised reader and might need more clarification? It is poorly described
- 'Analysis of the literature shows that four of these proteins have homologs in other bacteria that interact with RNA confirming our data.' Nice! But I think this would make a figure to give more confidence in the data? How many proteins have homologs and are detected as RBP in other bacteria but not here?
- 'Given that Y84 represents an RNA-binding site, we hypothesise that the aromatic ring of the tyrosine residue is important for the interaction with RNA.' This is such a simple experiment (and detailed above) and would clarify some mechanisms/make more impact!
- What is the link between *speB* expression and *yebC* knockout given that YebC-dependent pausing did not affect *speB* nor its regulator- maybe it's all indirect? No mechanism shown here.
- I would have liked to see more comparisons done for YebC with EF-P and YfmR – i.e. structural similarities
- The last paragraph was tagged on after the manuscript had been written – its contents should be better incorporated into the rest of the manuscript.

Overall, the manuscript presents an important contribution to the field of bacterial RBPomes, with well-executed experiments and clear data presentation. However, the novelty of the findings is somewhat undermined by existing studies on YebC, and the interpretation of the data would benefit from deeper integration with other RBPome datasets. Addressing the concerns regarding experimental rigour and expanding on comparative analysis will significantly enhance the manuscript's impact.

Reviewer #5

(Remarks to the Author)

Version 2:

Reviewer comments:

Reviewer #1

(Remarks to the Author)

The authors have satisfactorily responded to my previous comments and made the necessary changes to the manuscript.

Reviewer #3

(Remarks to the Author)

One minor change necessary – in lines 68 and 382 please change *B. subtilis* YebC to YebC2 (Yeel). The YebC (YrbC) protein of *B. subtilis* has not yet been shown to have a role in translation. I have no other concerns about the revised manuscript.

Reviewer #4

(Remarks to the Author)

We are pleased to see that the authors have addressed the majority of our concerns and suggestions. We are happy therefore to recommend publication of this manuscript in its revised form

Reviewer #5

(Remarks to the Author)

REVIEWER COMMENTS

Reviewer #1 (Remarks to the Author):

The manuscript by D. Ignatov and colleagues reports an approach developed for the identification of RNA binding proteins (RBPs) and experiments that support for the characterization of the newly identified RBPs. The authors provide compelling evidence that their approach successfully identified a number of proteins that interact with RNA in *S. pyogenes*. In the second half of the manuscript, the authors focused on one of the identified proteins, YebC. Deletion of *yebC* reduced the expression of the major *S. pyogenes* virulence factor, *SpeB*, and survival in human macrophages. Global mapping of the binding sites for YebC-3xFLAG suggested that YebC is a translational factor, and using the ribosome profiling as well as in vivo and in vitro translational reporter assays, YebC were demonstrated to promote the efficient translation of proteins containing polyproline stretches. This study nicely combines global analyses and biochemical assays, providing a convincing case for YebC's function for the efficient translation of proline-rich regions that will be of interest to researchers studying translation mechanism. The authors describe the methods in sufficient detail. A couple of points that are listed below seem important to further clarify, which the manuscript could be strengthened.

We thank the reviewer for their thoughtful and constructive comments, as well as for their positive evaluation of our work.

1. In Fig. 4C, the authors only show the secondary structure of part of 23S rRNA in order to map the location of the crosslinking region with YebC. There are structures of the ribosome available, and the peptidyl transferase center seem conserved among bacterial ribosomes. Hence, one of the ribosome structures should be used for the discussion to get better idea of the action of YebC in the translation of proline-rich regions.

We thank the reviewer for the suggestion. We added the structure of *E. coli* 23S rRNA with the cross-linking region highlighted as Figure 5D.

2. In Fig. 6C, the decrease in the whole-proteome level of the *yebC* mutant was slight (statically significant in the log phase). In this proteome analysis, did the authors use the information of full-length proteins? Given the YebC affects translation elongation at certain positions, the comparison of N-terminal and C-terminal portions in each protein would help in accurately understanding the contribution of YebC.

We performed the additional analysis of the MS proteomics data. The ribosome stalling at the PPP stretch of ValS protein was particularly strong. We compared the intensities of peptides located upstream and downstream of the PPP motif in the *WT* and $\Delta yebC$ strains, but did not detect any statistically significant difference (results can be provided on request).

3. In Supplementary Fig 7, interaction of YebC with the ribosome was not observed. In Fig. 6D, an in vitro translation system clearly demonstrated the YebC contribution to the translation of proline-rich regions. This pure condition would be appropriate to investigate the interaction of YebC with the ribosome. Can the authors show the polysome profile (or pull-down by YebC-3xFLAG) of the in vitro translation complex on the P5 reporter mRNA?

We wanted to detect a condition, where YebC associates with the ribosome that is stalled at the polyproline motif. According to our results, YebC_I resolves the stalling more efficiently than YebC_II and the complex of YebC_I with the ribosome might be short-lived. Therefore, we

performed the *in vitro* translation reaction containing the *E. coli* YeeN (YebC_II) for a short time and resolved the reaction on sucrose gradient. The peaks corresponding to 70S and polysomes were detected (lines 290-292 and Supplementary Fig. 9B). However, western blotting of the sucrose gradient fractions did not detect any association of YebC_II with ribosomes.

4. In discussion (lines 400-412, 432-442), the authors point out differences in mechanisms for YebC and EF-P, but they also refer previous studies that showed phenotypes caused by YebC, some of which are likely relevant to the EF-P function. Does the *S. pyogenes* genome have the *efp* gene? If this is the case, did the authors investigate phenotypes, such as growth curve (Supplementary Fig 5A), of an *efp* mutant or a *yebC efp* double mutant?

We thank the reviewer for this suggestion. *S. pyogenes* encodes the *efp* gene, but for reasons that are not entirely clear, we were unable to obtain a strain of *S. pyogenes* with the *efp* gene deleted. We therefore carried out this experiment in *Salmonella* Typhimurium, where an *efp* deletion causes mild growth attenuation.

The genome of *S. Typhimurium* encodes two paralogs of *yebC*: *yebC* (*yebC_I*) and *yeeN* (*yebC_II*). The $\Delta efp \Delta yeeN$ strain grew similarly to the Δefp strain (Fig. 8B; lines 862-863). In contrast, we were unable to generate the $\Delta efp \Delta yebC$ strain. The depletion of *yebC* in the Δefp strain performed with CRISPRi caused significant growth attenuation (Fig. 8C; lines 864-870). The observed growth attenuation suggests functional redundancy between EF-P and YebC. Further, these data corroborate the results of the *in vitro* translation assay and suggest that YebC_I might be more potent than YebC_II in resolving the ribosome stalling at the polyproline stretches (lines 294-308).

5. Related to Q4, in Fig. 2D the authors showed the phylogenetic distribution of YebC family proteins among bacterial genomes. To compare this with that of EF-P (or YfmR) may help in understanding the roles of these translational factors in the translation.

Our preliminary analysis showed that many bacterial species possess all three proteins. However, an in-depth analysis of their distribution would require a significant amount of bioinformatics analysis, which we feel is beyond the scope of our study.

6. In Supplementary Fig 5A, the *yebC* deletion increased the cell growth at the stationary phase under chemically defined medium condition, but not under the other medium conditions. Did the authors compare the expression level of *yebC* under these culture conditions?

We appreciate the reviewer's suggestion. We studied expression level of YebC in THY and CDM using western blotting and it appears to be similar (lines 141-142 and Supplementary Fig. 5A).

7. In lines 446-448, the authors referred that the expression of *SpeB* is affected by a quorum sensing and the environmental pH. In the transcriptome and proteome analyses of the *yebC* mutant, were the factors that sense these environmental cues separately investigated? Does "its regulator" in line 447 indicate the factor involved in quorum sensing or pH modulation?

We performed additional experiments to understand the reasons for *speB* downregulation in the $\Delta yebC$ strain (lines 145-150; Fig. 3B and 3C). Our results demonstrate that translation and secretion of *SpeB* are not affected by *yebC* deletion. In contrast, the decreased activity of the *speB* promoter seems to be the reason for the downregulation of *speB* expression in the $\Delta yebC$ strain. The *speB* promoter is regulated by a quorum-sensing pathway consisting of a secreted leaderless peptide signal (SIP), and its cognate receptor RopB. Acidification of the environment increases

affinity of RopB for the leaderless peptide and serves as a second factor increasing SpeB expression. This clarification has been added to the discussion (lines 362-368).

Reviewer #2 (Remarks to the Author):

In the manuscript entitled "Novel RNA-binding protein YebC enhances translation of proline-rich amino acid stretches in bacteria" submitted to Nature Communications, Ignatov and colleagues sought to identify and investigate novel RNA binding proteins in group A Streptococcus, an important human pathogen. Using the global approaches OOPS and RBS-ID, the authors discovered a set of previously unreported, putative RNA binding proteins, in addition to established ones (Fig. 1 and Table 1) in *S. pyogenes*. The authors initially focused their efforts on five putative RNA binding proteins assessing their ability to interact with RNA in vivo and found via the immunoprecipitation of crosslinked RNA complex approach that YebC, PhoH, ThuC, YjbK and YgaC interact with RNA (Fig. 2). The authors then concentrated on YebC, which is highly conserved in bacteria and eukaryotes, but little was known regarding its function until very recently. Through deletion and mutation analyses of *yebC*, the authors provide evidence that YebC is important for *S. pyogenes* survival in macrophages and expression of the virulence factor SpeB. The authors also established the role of particular domains and residues in YebC function (Fig. 3). Using an exhaustive set of high throughput, molecular, and biochemical approaches, the authors determined that YebC interacts with the 23S rRNA and assists the ribosome in translating through stretches of regions of mRNA encoding multiple proline residues (Figs. 4, 5, and 6).

Overall, this manuscript is well written and details a thorough investigation of RNA binding proteins in *S. pyogenes* and in particular YebC, establishing a detailed molecular mechanism of the function of this protein that is highly conserved in at least two domains of life. The overwhelming amount of data presented by the authors resolutely support their model of YebC function. While there is a manuscript recently published in NAR that identified the human homologue TACO1 in assisting the ribosome in translating through proline rich codons, it is my opinion that this work only slightly detracts from this manuscript under review. This manuscript not only validates the conclusions of that work by Brischigliaro et al (PMID 39036954), but also establishes additional details of the molecular mechanism of YebC demonstrating that it interacts with a specific region of the 23S rRNA, establishing the role of distinct domains and residues in its function, and demonstrating the impact of this protein on *S. pyogenes* in terms of its growth, virulence, and proteome. Additionally, the OOPS and RBS ID data generated from this work revealed many other putative RNA binding proteins, and this data is useful for others investigating RNA binding proteins and RNA metabolism in both closely and distantly related bacteria that have homologous proteins. In summary, this is an exciting, highly impactful manuscript.

We are grateful to the reviewer for their careful assessment of our work and for their constructive and supportive comments, which recognize the relevance and potential impact of our findings.

Major Criticisms:

NONE

Minor comments:

Fig. 4A legend. Although it is defined in the methods section, it would have been useful to me and perhaps others, if the authors defined SMI here as size matched input.

We have removed the abbreviation SMI from the Results section of the manuscript and only use it in the Methods section.

Fig. 4D. Since I and perhaps other readers are not familiar at looking at traces of ribosomal complexes, it might be helpful to provide some labeled axis for these results such as labeling an X-axis as sucrose concentration and Y-axis as signal intensity, unless some other convention exists.

We labelled the X-axis as "10-50% sucrose gradient" and the Y-axis as "OD₂₆₀".

References. The names of species are not italicized as is typical convention.

We thank the review for pointing that out. We have corrected the references accordingly.

Reviewer #3 (Remarks to the Author):

The manuscript by Ignatov et al. describes the identification and characterization of YebC_{II} as a translation factor that facilitates translation of polyprolines in *Streptococcus pyogenes*. The authors identified YebC_{II} as an RNA binding protein following UV-cross linking and mass spectrometry. Indeed, this screen uncovered a number of novel RNA-binding proteins as well as re-confirming other known-RNA-binding proteins. Authors were also able to identify residues that are likely RNA-interacting on YebC as well as the region of the ribosome (Helix89 near the PTC) with which YebC is likely to interact. These data are very helpful in lieu of a structure. Authors follow up on the YebC residues of interest with site-directed mutagenesis. Finally, authors performed ribosome profiling to identify mRNA sequences with significant pause scores and found stalling at PP, PXP, or DXP motifs. Authors followed up using polyproline-encoding in vitro and in vivo reporters.

The study is thorough, well-executed, and very timely. Major claims are well supported by the data. Authors present a comprehensive mix of proteomics, biochemistry, genetics, ribosome profiling, and specific reporters to support their findings.

Specific comments for improvement:

We thank the reviewer for their careful evaluation of our manuscript and for their positive and encouraging comments regarding the thoroughness and significance of our study.

1. Line 57: Authors should re-word the statement "YfmR/Uup has also been shown to promote translation of proline-rich motifs in cooperation with EF-P"

This wording makes it sound like EF-P and YfmR work together to relieve proline stalling, but this is unlikely to be the case since binding of one would hinder binding of the other. Deletion of both factors indeed causes a severe growth defect with many orders of magnitude decreased survival, but all current evidence suggests EF-P and YfmR function independently on the ribosome.

We have re-worded the statement (lines 64-65).

2. Line 125: Authors should modify language surrounding the conservation of YebC. It is not clear that YebC is found in most bacteria. To say that the protein is broadly conserved would be more accurate, especially since the degree of conservation is different for each paralog. It is also not accurate to say it is universally conserved without the identification of an Archaeal homolog.

We have modified the statement using the term "widely conserved" (line 19).

3. The identification and validation of residues involved in RNA-binding is very exciting. Can the presence or absence of these residues in the YebC_I family be used to speculate on their function? Are they conserved in YebC_I or absent? Either finding would be interesting since one exciting mystery is whether the YebC_I family proteins may also function in translation. A modest role for YebC_I in translation is also supported by Figure 6D, though it is not as significant as for YebC_II. Marking the important YebC_I residues in Figure S4 would be helpful.

In the revised manuscript, we have re-named the RNA-binding sites to the “RNA cross-linking sites”, because the identification of cross-linking to a particular amino acid does not prove that the amino acid specifically interacts with RNA. Nevertheless, we have demonstrated the importance of amino acids forming the positive patches on the YebC surface (M2) for the activity of the protein. We have labelled these amino acids in Supplementary Fig. 4: most of them are conserved in both YebC_I and YebC_II variants. An invariant Y84 was also shown to be important for the protein activity. In the revised manuscript, we also present the results of the OOPS experiment demonstrating that both M2 and Y84 are important for the interaction with RNA (lines 180-182 and Fig. 4D).

The *in vitro* translation experiment demonstrated that both YebC_I and YebC_II versions from *E. coli* resolve the ribosome stalling at the polyproline motif (Fig. 8A). In the revised manuscript we also demonstrate a partial redundancy between *efp* and *yebC_I*, but not *yebC_II*, in *Salmonella* Typhimurium (lines 294-308; Fig. 8B and 8C and Supplementary Fig. 13 and 14). These results and the high degree of sequence conservation suggest that the two versions of YebC perform similar functions. Although this requires further confirmation, we suggest that there is some degree of functional diversification for the YebC versions (lines 374-380).

4. Authors should reduce claim that EF-P and YfmR are absent in the PURE system. The PURE system certainly has a lot of additional ribosome-associating factors that co-purify with ribosomes. I do not find this problematic for the conclusions of the study, just an important thing to recognize about PURExpress, and particularly important because of the potential for YebC to bind the A-site. If that is the case, EF-P and YfmR/Uup could still bind the ribosome at the same time even if they aren't necessarily working together.

We have removed the statements that the PURE system does not contain EF-P and YfmR (lines 274-275).

5. I am curious about increased optical density in stationary phase. What could be the reason for this? Have authors looked at mutant cells under the microscope? Does the cell volume increase in stationary phase?

The reason for the increase in optical density is unclear. We checked the cells under the microscope and measured their volume (line 143; Supplementary Fig. 5B and 5C). The *yebC* deletion does not affect the cell volume. Since the *yebC* deletion affects gene expression in a pleiotropic way, it could be difficult to understand what causes this phenotype.

6. The differential expression of *speB* remains mysterious. Although a significant pause-site was not identified in the ribo-seq data, are there are other features of *speB* that could be of interest for ribosome stalling? For example, prolines near the 5' or 3' end where the stalling may be less obvious due to increased ribosome occupancy from initiation or termination? Are there transcriptional regulators upstream of *speB* that exhibit ribosome stalling? It would be interesting

to compare an in vivo and an in vitro *speB* reporter but not essential for the conclusions of the paper.

We performed additional experiments to study the expression of *speB* (lines 145-150; Fig. 3B and 3C). When SpeB was expressed from the heterologous *Ptet* promoter, its expression was not affected by the *yebC* deletion. This shows that the translation and secretion of *speB* do not depend on *yebC*. On the other hand, the fusion of the *speB* promoter with sfGFP demonstrated that it is the promoter activity that is affected by the *yebC* deletion. The *speB* promoter is regulated by a quorum-sensing pathway consisting of a secreted leaderless peptide signal and its cognate receptor RopB. Acidification of the environment increases the affinity of RopB for the leaderless peptide and serves as a second factor in increasing SpeB expression. RopB does not have the polyproline amino acid stretches, so the observed downregulation of *speB* expression is likely a secondary effect of the *yebC* deletion (lines 362-368).

Minor:

Typo at line 21 – missing “an” before “in vitro system”.

This has been corrected (line 22).

Reviewer #4 (Remarks to the Author):

Review of Ignatov et al., “Novel RNA-binding protein YebC enhances translation of proline-rich amino acid stretches in bacteria”

The manuscript from Ignatov et al, entitled “Novel RNA-binding protein YebC enhances translation of proline-rich amino acid stretches in bacteria” describes a comprehensive proteome-wide study in *S. pyogenes*, to retrieve its RNA binding proteome (RBPome). The authors took two approaches to enrich the RBPome: OOPS, an approach using phase separation of UV crosslinked RNA-bound proteins, and RBS-ID, a method that uses hydrofluoride to fully cleave RNA into mono-nucleosides and allows RNA binding site mapping at single amino acids resolution. Using these approaches the authors were able to return a credible list of RBPs from *S.pyogenes*. They went on to delve into the function of one of these proteins, YebC, their data suggesting a role in facilitating translation at sites of polyproline in translation proteins.

This work is timely, insomuch that many large scale RBPome studies are available, but few go onto the interrogate the function of ‘novel’ RBPs present in their data. The study is mostly well executed, but lacks a certain novelty, as other studies have already assigned function to YebC in other organisms, and thus the main results of the study are not surprising. However, the data presented do represent the only RBPome data from this pathogenically important bacterium.

The manuscript does require some clarification for it to be suitable for publication. Moreover, a fuller description of existing publicly available datasets is also required to provide an up-to-date representation of the field.

We are grateful to the reviewer for their careful review and constructive feedback, which have helped us to further clarify and strengthen the manuscript.

Major Comments:

1. Novelty/contribution to the field: The authors have done a commendable job in identifying RBPs in *S. pyogenes*, but YebC has already been characterised in *E. coli* (Monti et al., 2024) and in human studies (as TACO1). This weakens the novelty of the central finding. The introduction would

benefit from acknowledging these studies earlier, providing a more cohesive backdrop to YebC's conserved functions across species and why the authors focused on this protein.

We acknowledge the recent publication regarding TACO1. Our manuscript not only corroborates the findings of the aforementioned paper but also provides insights into the molecular mechanism of YebC. Moreover, we characterized the physiological effects of this protein on the growth and virulence of *S. pyogenes*. Additionally, we identified numerous potential RNA-binding proteins in this pathogenic bacterium, which will be available for examination by other researchers in the field.

We have re-written the introduction to mention the most important studies on the YebC protein (lines 52-72).

2. Dataset Comparisons: The study's claim of novelty in identifying YebC and other RBPs would be stronger if the authors systematically compared their findings with other RBPome studies. Expanding the supplementary data to include orthology-based comparisons across species (including human and *E. coli* RBPomes) would significantly enhance the robustness of the findings.

We compared our results with five other studies that identified RBPs on a whole-proteome scale in bacteria:

1. *E. coli*: Shchepachev, V., et al. (2019) *Mol Syst Biol* 15(4): e8689.
2. *E. coli*: Queiroz, R. M. L., et al. (2019) *Nat Biotechnol* 37(2): 169-178.
3. *E. coli*: Monti, M., et al. (2024) *Mol Syst Biol* 20(5): 573-589.
4. *E. coli*: Stenum, T. S., et al. (2023) *Nucleic Acids Res* 51(9): 4572-4587.
5. *S. Typhimurium*: Urdaneta, E. C., et al. (2019) *Nat Commun* 10(1): 990.
6. *S. aureus*: Urdaneta, E. C., et al. (2019) *Nat Commun* 10(1): 990.

We did not include the eukaryotic studies because of the large differences in the physiology between eukaryotes and prokaryotes. The results of the comparisons are provided in lines 98-99, Table 1, Supplementary Table 1 and Supplementary Fig. 3C and 3D. The orthologs of 13 *S. pyogenes* RBP candidates have previously been classified as RBPs in other bacterial species.

3. Methodological points:

○ The UV crosslinking dose (600 mJ/cm²) used in the OOPS method is not justified in the manuscript. The authors should provide evidence of a dose-response experiment to support this choice, particularly as they are using a non-model organism or at least disclose why they chose this dose.

We performed a titration of UV-C energies using the YhaM and GapN proteins as a positive and a negative probe respectively. Out of 300, 600 and 1200 mJ/cm², the energy of 600 mJ/cm² resulted in the highest enrichment of YhaM relative to GapN (Supplementary Fig. 1B).

○ Clarification on the use/limitations of SILAC in *S. pyogenes* is required, as this bacterium synthesises its own amino acids, which may limit efficient isotope incorporation.

S. pyogenes is a natural auxotroph for lysine and arginine (Gera, K. and K. S. McIver (2013). *Curr Protoc Microbiol* 30: Unit 9D 2.). We have added this explanation to the text (line 82).

○ Figure 2B would strongly benefit/would be more convincing if they did a densitometry quantification between the UV+PNK+ vs UV-PNK+ at the minimum. The negative control (enrichment 0.01) looks like it has a stronger signal/shift compared to YgaC (enrichment 3.21) ...

this might just be because it needs to be normalised by the protein amount in the WB. A barplot will make the values much more intuitive/the results more convincing/conclusive.

In our experience and according to the published studies (see Urdaneta, E. C., et al. (2019) Nat Commun 10(1): 990 as an example), the PNK assay is a qualitative rather than a quantitative method. Due to the imprecise nature of densitometry, normalization of the radioactive signal to the intensity of the western blotting bands would significantly bias the result.

in our opinion, the visual comparison of the radioactive bands in "UV+ PNK+" and "UV- PNK+" provides a better estimate of how efficiently the protein cross-links with RNA. Whereas for GapN, the "UV+ PNK+" radioactive band is only slightly stronger than the "UV- PNK+", for YgaC the "UV- PNK+" band is not visible at all.

4. OOPS and RBS-ID Analysis:

- The manuscript lacks a detailed interpretation of the physicochemical properties of the amino acids enriched in the RBS-ID experiment. Do the enriched sites match known RNA crosslinking motifs? Including this analysis would provide more confidence in the approach.

These results would require a complex bioinformatic analysis. Since our study focuses mostly on the characterization of YebC, we believe that this analysis is beyond the scope of our work.

- GO term analysis is insufficiently detailed, especially for bacterial systems. Integrating orthology information to identify conserved RBPs across species would be more meaningful than relying solely on GO term enrichment, which is limited for bacterial systems.

We agree that the GO term analysis is not sufficiently detailed, particularly for non-model bacteria such as *S. pyogenes*. The identification of conserved RBPs is described in response to the major comment #2 (lines 98-99, Table 1, Supplementary Table 1 and Supplementary Fig. 3C and 3D).

- 'The annotated RBPs, the proteins with significant OOPS enrichment, and the 87 proteins with detected RNA-binding sites overlapped significantly (Fig. 1B)'. Figure 1B requires clarification of the significance of the overlap between different datasets (OOPS, annotated RBPs, and RBS-ID proteins). A statistical test (e.g., a contingency table) to quantify this overlap would be useful.

Thanks for pointing this out. This sentence is indeed imprecise. To avoid too much detail, we have decided to remove it.

5. Clarification of Terminology:

- The authors' use of the term "novel RBPs" is questionable. Many proteins described as novel RBPs in this manuscript have been identified in other bacterial species, even if uncharacterized. It would be more accurate to describe them as "uncharacterized" rather than "novel." (or some other term that the authors find more representative).

We believe that the classification of a protein as an RBP using the MS proteomics approaches such as OOPS is not a sufficient criterion for considering a protein as a genuine RBP. First, it might be possible that these approaches false-positively identify some proteins as RBPs. Second, cross-linking to RNA does not guarantee that the protein specifically interacts with RNA and can be explained for example by the interaction with another RBP leading to proximity to RNA molecules. This is why we re-named these proteins "Candidates for novel RBPs" or "RBP candidates" in our manuscript.

- The interaction sites of YebC should be contextualised with respect to known RNA-binding domains (e.g., helix-turn-helix domains) to better link RBS-ID data with biological function.

Although several studies suggest that YebC contains the helix-turn-helix domain, this domain has not been annotated in the UniProt or any other database. Therefore, we do not have sufficient evidence that YebC contains the helix-turn-helix domain.

Minor Comments:

- The introduction does not properly contextualise the field of RBPome in the bacterial world and/or explain why it has been delayed compared to the eukaryotes due to experimental limitations. "These studies have identified ProQ" is a very oversimplified summary of the yield of RBPome studies in this area. It would be to the benefit of the papers for the authors to highlight the relatively few manuscripts (in bacteria) which have functionally dissected the role of newly-identified RBPs. This paper is part of a larger community effort to start characterising the RBPome beyond the accumulation of catalogues, but the value of this needs to be made more clear in the introduction. Moreover, Monti et al also discusses YebC as a novel E. coli RBP and describe a mitochondrially localised human orthologue – they should acknowledge this in the introduction rather than leave it to the discussion. Finally, in the discussion they mention a paper in humans demonstrating the role of TACO1 (YebC ortholog) in resolving polyP in mitochondria, but this was not mentioned in the introduction.

The introduction section of the experimental articles should be necessarily concise. As our article focuses mostly on the characterization of YebC, we believe that we provide sufficient coverage of the RBPome field in bacteria.

We have significantly expanded the discussion of YebC in the introduction and mentioned the paper by Monti et al., as it has been suggested by the reviewer (lines 52-53). However, we should acknowledge that the homology between YebC and TACO1 was first mentioned in the study by Zhang, Y., et al. (2012). BMC Syst Biol 6 Suppl 1: S20.

- The supplementary tables are excellently labelled and very easy to understand/navigate, we thank the authors.

We appreciate the positive feedback!

- We would recommend expanding Supplementary data 2 to include orthology-inferred RBPs- it's high time we start integrating all the RBP catalogues to get a more refined picture of what is going on instead of collections of stand-alone catalogues of 'novel' RBPs- I wouldn't be surprised if many of these 'novel' candidates actually appear to be consistently identified as RBP in other species. This would be significant and make the data more robust!

We have included this information to the Supplementary table 1.

- On this note Supplementary figure 2 would be extremely informative to the field if it incorporated the RBP info from human, mouse, ... Ecoli to see which candidates are constantly appearing.

Since our paper is mostly focused on the characterization of YebC, we have decided to remove the discussion of glycolysis enzymes from the manuscript. We believe this will make the manuscript less complex to read.

- Further, the red and blue dots in Sup. Fig. 2 are not clear and are not labelled in the caption- they must be the OOPS enrichment and the detection of RNA-binding, but which is which?

Supplementary Fig. 2 has been removed from the revised manuscript.

- Additionally- would be intrigued to see which GO terms are found in the different categories of Fig 1B, e.g. are the proteins pulled out in RBS-ID only different functionally/physicochemically compared to those only seen in OOPS?

Our study focuses on the characterization of YebC. Therefore, we believe that the validation of OOPS and RBS-ID is beyond the scope of our study.

- 'Glycolysis enzymes from different species often function as non-canonical RBPs' I would suggest expanding the citations to primary research/RBPome studies that demonstrate and validate this clause instead of a review.

The discussion of glycolysis has been removed from the revised manuscript.

- Why did the authors not follow up YjbK, YgaC and ThuC ? In the discussion there is some description of what the protein might do, but the reason not to follow these up further could have been better enunciated.

We considered YebC as the most promising RBP candidate for further characterization.

- 9 of these are detected as DNA-binding proteins- need to evaluate in the light of recent findings from Chu et al 2022 (<https://doi.org/10.1038/s41467-022-30553-8>) where they demonstrate that DNA-binding and Rossmann-fold proteins show RNA-binding. Are these domains detected in these proteins? I think a quick look to see if this domain hypothesis is true would be meaningful.

We apologize, there was an error in the annotation: only 7 of the candidate RBPs are DNA-binding. The corresponding changes have been made in the manuscript. Four out of seven DNA-binding proteins contain the HTH domain. In two of them, the cross-linking sites are located in the HTH domain, and in the other two, they are outside of it (the results of the analysis can be provided by request). In conclusion, we did not observe any association of the RNA cross-linking sites with the HTH domains.

- Plot 2C is excellent- it makes me wonder- are the RBS-ID sites enriched in RNA-binding domains? It would be nice to see this analysis in the supplementary (!) and give more confidence to the approach.

We calculated the number of RNA cross-linking sites mapped to different RNA-binding domains and domains of the ribosomal proteins. The results are presented in the Supplementary Fig. 2C and 2D. As expected, the RNA cross-linking sites are often located in the domains of ribosomal proteins and the RNA-binding domains.

- ‘Suggesting that it plays an important role in bacterial physiology’ - no reference to the fact that in Monti et al 2024 the yebC KO the growth rate and saturation OD count was affected in both M9LB and under nutrient stress, perhaps a hypothesis to the matter can be explored in the discussion.

In *S. pyogenes* and *S. Typhimurium*, the deletion of *yebC* alone in the WT background does not significantly attenuate growth (Fig. 3A and Fig. 8B). Therefore, the *E. coli* data presented in Monti et al. could reflect the species-specific role of YebC in bacterial physiology.

- The human ortholog (TACO1) has been detected as RNA-binding in human assays (Queiroz et al, 2019; Castello et al, 2016 - not sure about mice!) but has not been referred to here.

Since TACO1 has been shown to participate in translation, it is highly likely that it interacts with ribosomal RNA. Therefore, the detection of its interaction with RNA in these studies is not of high importance for our research.

- Figure 2D: what is blue and red? Simply the species? So many of the species listed have an RBP dataset!! Please look, see if RNA-binding conserved, would make the conclusions so much more powerful!

We have coloured all the circles in grey. The YebC protein has indeed been classified as a novel RBP in several studies, which is indicated in Table 1.

- Figure 2E: Also note that YebC has a HTH domain (at least in E Coli) which as I mentioned above has been proven to bind RNA in vivo (Chu et al, 2022)- are the interaction sites in this domain? This would showcase the value of having both OOPS data AND RBS-ID to integrate and extract more meaningful/complete data.

We do not see evidence of the presence of an HTH domain in YebC.

- Why a C-terminal tag? Is this further (structurally) from the predicted binding region of the protein?

The N-terminal tag is located close to the ribosome-binding site and can potentially affect the level of protein translation. Additionally, N-terminal FLAG could co-IP ribosomes that are actively translating our proteins under study. To avoid this, we decided to perform C-terminal tagging.

- Describe what SpeB is when it is first mentioned.

When SpeB is mentioned for the first time, we write “the major virulence factor of *S. pyogenes*, the secreted protease SpeB” (lines 145-147). Since the study does not focus on *S. pyogenes* virulence, we believe that this is sufficient. In the Discussion we provide more information on the regulation of SpeB expression (lines 364-367).

- Confusing that Figure 3A is the same as Sup. Fig. 5C.

We have removed Figure 3A.

- Do the mutations affect the RNA-binding ability? Could the authors take the yebC-/yebC+ variants for each mutant, apply OOPS and evaluate the WB signal of YebC in the organic vs the interphase? This would be essential to decouple: is the capacity to bind RNA necessary for YebC function? Or instead, is RNA binding an additional function of YebC?

We thank the reviewer for this suggestion. We used OOPS to probe the amount of the WT YebC and its mutants, M2 and Y84A, in the RBP-enriched fraction (lines 180-182; Fig. 4D). The results show that the mutations disrupt the interaction with RNA. Since these mutations also make the protein unable to resolve the ribosome stalling at the polyproline motifs, we suggest that the interaction with RNA is important for the activity of YebC (lines 332-334 and 337-338).

- In the results, a clearer explanation of the iCLIP data analysis, especially for non-expert readers, is necessary. Terms like "size-matched input libraries" should be defined, and a better explanation of the rationale behind 'we extracted the cross-linked nucleotides located immediately upstream of the mapped cDNA reads'.

We have improved the description of the iCLIP experiment (lines 193-198).

- I am not sure they included the right control for iCLIP- you are irradiating the cell so RNA that is proximal to YebC will be covalently bound. Given the sheer abundance of ribosomes, there is a chance you are capturing noise... I think the most elegant would have been to do the experiment I suggest above of seeing which mutants are unable to bind RNA and use that 'RNA-binding-dead' mutant as a control. It could be that YebC interacts with the ribosome not necessarily the 23S rRNA. In any case this could be addressed with an EMSA of the YebC and the RNA-binding-dead YebC variant to avoid the cost of repeating iCLIP. This of course would only be the case if the authors obtain reproducible/convincing EMSA shifts already with YebC in WT state.

We are aware of the problem with the ribosome abundance, and therefore used three negative control libraries in our iCLIP experiment. We are not sure that EMSA would work for YebC, since this protein interacts with the whole ribosome in a transient manner.

- 'Our search for RBPs in *S. pyogenes* identified most of the proteins involved in translation, including ribosomal proteins, proteins involved in ribosome biogenesis and translation factors. Therefore, it seems possible that YebC is also involved in ribosome biogenesis or translation. In order to investigate' - I do not see the immediate connection between these sentences here, need to be more explicit. Most RBPs will be translation related because those are the best annotated and most conserved (!) and possibly the easiest to capture due to their abundance and near constant binding to RNA.

We have removed this statement.

- The distinction between the use of different media (THY for RNA-seq and CDM for KO characterization) needs to be clarified to avoid confusion.

For each experiment, we indicated the media for cell growth in the Results and Methods sections. We used CDM for OOPS because it allows to perform SILAC labelling. We also used CDM for iCLIP, because the UV-C energy has been titrated for cells grown in this medium. Since the $\Delta yebC$ strain grew differently in CDM, we used this medium for the phenotype characterization. For all other

S. pyogenes experiments, we used THY medium. We do not always provide the rationale for the choice of medium to facilitate reading of the Results section.

- More emphasis on comparing the effects of YebC deletion with other known factors such as EF-P and YfmR would enhance the discussion of potential mechanisms underlying ribosome stalling.

The mechanism of ribosome stalling at the polyproline motifs has been studied in Huter P, et al. *Mol Cell* 68, 515-527 e516 (2017). The same study also describes a mechanism of the stalling resolution by EF-P. For YfmR and YebC this mechanism is unclear. In the discussion, we compare the ribosome profiling results of the $\Delta yebC$ strain in our study and Δefp strains in the studies by Hummels et al. *PLoS Genet* 15, e1008179 (2019) and Woolstenhulme et al. *Cell Rep* 11, 13-21 (2015). There are some differences in the ribosome pausing that are discussed in lines 350-354.

- Supplementary fig. 8A, why only show the length distribution of one library? Can the authors comment on the distribution shown?

For the other libraries, the length distribution looks very similar. This length distribution demonstrates that we used the appropriate concentration of S7 Nuclease to partially degrade mRNA: most of the reads have a length between 27 and 40 nt, which corresponds to the length of the ribosome footprints, as shown by previous ribosome profiling experiments in bacteria.

- 'Median pause scores were slightly higher when serine or glycine codons were located in the E site (Supplementary fig. 8C)' which statistics have been used, if any? Please clarify.

This effect is a well-known artefact of the ribosome profiling experiments (see Mohammad, F., et al. (2019). "A systematically-revised ribosome profiling method for bacteria reveals pauses at single-codon resolution." *Elife* 8). Since the observation is not new, we did not perform a thorough statistical analysis. What is more important here, is that *yebC* deletion does not affect the pausing in the amino acid-dependent manner.

- Unclear what the x-axis of 5B is!

The x-axis shows the position of the paused ribosome with the P and A sites indicated. This description has been added to the legend of Fig. 6B.

- 'To this end, we performed a proteome analysis of the same bacterial cultures that we used for the transcriptome analysis: WT, $\Delta yebC$ and $\Delta yebC$ / *yebC*⁺ strains in mid-logarithmic and stationary growth phases (Supplementary data 7)' - missing to state the conclusion, where there any changes overall in the proteome in these pairwise comparisons? Visualising this in the supplementary. Would perhaps expect some changes if the phenotype is ribosome activity?

The proteomes of the strains were very similar, suggesting that the *yebC* deletion affects the translation very slightly: very few genes were differentially expressed between the WT and $\Delta yebC$ strains (lines 267-269 and Supplementary Fig. 12). We have added this text and the supplementary figure to the manuscript.

- For later in the discussion but why is 'ns' in the stationary phase in Fig 5C?

'ns' denotes 'not significant'. The explanation has been added to the legend for Fig. 7B.

- An additional thought- is there any functional relevance to genes with polyP in this system? Would be interesting to do GO/KEGG on the candidate's pauses (and genes with polyP generally). If no terms appear significant, I would explore applying this on the Ecoli orthologs, as they may be better annotated. Perhaps a sliding approach is needed P1 -> P3 -> P5 given the changing significance depending on the extension of the polyP track shown in Fig 6.

We believe that the proposed experiment is beyond the scope of the study. The distribution of the polyproline motifs in bacteria has been studied by Starosta, A. L., et al. (2014). Cell Rep 9(2): 476-483.

- 'We next used the PURE in vitro translation system' - after reading the discussion I understand this was done to exclude indirect effects of EF-P or YfmR but this is not immediately obvious to the non-expert reader. I would make the rationale more transparent already in the results section.

One of the reviewers has pointed out that PURE system might contain residual amount of EF-P and YfmR. Therefore, we have removed this statement from the manuscript.

- In the discussion: 'This significantly narrowed the set of candidates for novel RBPs and filtered out the proteins that could stably interact with polysaccharides' - I think the rationale for this concern (contaminating interphase) is not immediately obvious to the non-specialised reader and might need more clarification? It is poorly described

To make the text easier to understand, we have removed this statement.

- 'Analysis of the literature shows that four of these proteins have homologs in other bacteria that interact with RNA confirming our data.' Nice! But I think this would make a figure to give more confidence in the data? How many proteins have homologs and are detected as RBP in other bacteria but not here?

See our response to the Major comment #2.

- 'Given that Y84 represents an RNA-binding site, we hypothesise that the aromatic ring of the tyrosine residue is important for the interaction with RNA.' This is such a simple experiment (and detailed above) and would clarify some mechanisms/make more impact!

We present the results of the suggested OOPS experiment in lines 180-182 and Fig. 4D.

- What is the link between *speB* expression and *yebC* knockout given that YebC-dependent pausing did not affect *speB* nor its regulator- maybe it's all indirect? No mechanism shown here.

We performed additional experiments to study the expression of *speB* (lines 145-150; Fig. 3B and 3C). When *SpeB* was expressed from the heterologous *Ptet* promoter, its expression was not affected by the *yebC* deletion. This shows that the translation and secretion of *speB* do not depend on *yebC*. On the other hand, the fusion of the *speB* promoter with sfGFP demonstrated that it is the promoter activity that is affected by the deletion of *yebC*. The *speB* promoter is regulated by a

quorum-sensing pathway consisting of a secreted leaderless peptide signal and its cognate receptor RopB. Acidification of the environment increases the affinity of RopB for the leaderless peptide and serves as a second factor in increasing *SpeB* expression. RopB does not have the polyproline amino acid stretches, so the observed downregulation of *speB* expression is likely a secondary effect of the *yebC* deletion (lines 362-368).

- I would have liked to see more comparisons done for YebC with EF-P and YfmR – i.e. structural similarities.

The proteins do not share any common domains and their folds are completely different. Therefore, the structural alignment of these proteins would not make sense.

- The last paragraph was tagged on after the manuscript had been written – its contents should be better incorporated into the rest of the manuscript.

We have incorporated this text into the Introduction (lines 52-72).

Overall, the manuscript presents an important contribution to the field of bacterial RBPomes, with well-executed experiments and clear data presentation. However, the novelty of the findings is somewhat undermined by existing studies on YebC, and the interpretation of the data would benefit from deeper integration with other RBPome datasets. Addressing the concerns regarding experimental rigour and expanding on comparative analysis will significantly enhance the manuscript's impact.

REVIEWERS' COMMENTS

Reviewer #1 (Remarks to the Author):

The authors have satisfactorily responded to my previous comments and made the necessary changes to the manuscript.

Reviewer #3 (Remarks to the Author):

One minor change necessary – in lines 68 and 382 please change *B. subtilis* YebC to YebC2 (Yeel). The YebC (YrbC) protein of *B. subtilis* has not yet been shown to have a role in translation. I have no other concerns about the revised manuscript.

We introduced the changes suggested by Reviewer #3.

Reviewer #4 (Remarks to the Author):

We are pleased to see that the authors have addressed the majority of our concerns and suggestions.

We are happy therefore to recommend publication of this manuscript in its revised form

Reviewer #5 (Remarks to the Author):
